# Pressure-Relief Effect of Post-Op Shoes Depends on Correct Usage While Walking

**DOI:** 10.3390/bioengineering12050489

**Published:** 2025-05-02

**Authors:** Claudia Döhner, Christian Soost, Sam Steinhöfer, Jan A. Graw, Christopher Bliemel, Artur Barsumyan, Rene Burchard

**Affiliations:** 1Department of Medicine, Philipps-University of Marburg, 35037 Marburg, Germany; 2School of Economic Disciplines, University of Siegen, 57072 Siegen, Germany; 3Department of Anesthesiology and Intensive Care Medicine, Ulm University Hospital, 89081 Ulm, Germany; 4Department of Orthopaedics and Traumatology, University Hospital of Giessen and Marburg, 35032 Marburg, Germany; 5Department of Orthopaedics and Trauma Surgery, Lahn-Dill-Kliniken, 35683 Dillenburg, Germany

**Keywords:** forefoot relief, forefoot surgery, gait and posture, plantar pressure distribution, post-op shoe, motion analysis, quality of life, musculoskeletal disorders

## Abstract

Post-op shoes (POSs) are commonly used after forefoot surgery to protect the surgical site. However, there are insufficient data on their impact on forefoot load during the rollover phase of walking. This study aims to analyze the effects of a commonly used POS on plantar pressures under the forefoot and to assess whether improper usage could affect pressure patterns. Sixteen healthy volunteers underwent three different walking tests on a straight tartan track. The test setting included walking barefoot, as well as normal walking and a modified heel-accentuated “limping” gait while wearing a common POS. The pressure distribution over the forefoot regions of interest was measured using sensor insoles and a pressure-measuring plate on the ground. Results show that only the heel-accentuated “limping” gait in the POS led to a significant reduction in pressure values over all anatomical regions compared to the normal barefoot gait. Furthermore, higher pressure values were found over the lesser toes during normal walking in the POS compared to normal barefoot walking. The findings highlight that the protective function of a POS relies on proper use, specifically the correct gait pattern. If used incorrectly, POS may even have unfavorable effects on the pressure on the operated forefoot and possibly even increase the risk of delayed healing or complications in comparison to barefoot walking. Therefore, strategies such as patient training in proper walking techniques should be incorporated into postoperative care.

## 1. Introduction

Deformities of the foot may cause limitations in the physical and psychological quality of life of affected patients and can severely reduce their level of activity [1]. Deformities can occur in the hind- and midfoot as well as in the forefoot and specifically the toes [2]. Deformities in the toes are usually divided into those occurring in the first ray and those occurring in rays II to V [2]. Hallux valgus (HV) is one of the most prevalent foot disorders, affecting almost a third of the female population and worsening with age [3,4]. HV not only causes pain and difficulty finding suitable footwear but also impairs muscle activity and gait stability and could be associated with secondary deformities of the lesser toes [5,6,7]. Classic examples of these deformities are hammer toes or claw toes [8]. The treatment of both pathologies accounts for the most frequent surgical interventions on the forefoot [2,9]. The incidence of deformities of the second to fifth ray is unknown; however, population-based studies in Australia and Sweden report that surgical management of these deformities comprises between 28% and 46% of all forefoot surgery [10,11]. Available and state-of-the-art treatment options for toe deformities include conservative management and surgical techniques. More than 150 procedures are described for surgical treatment of forefoot deformities [12,13,14,15].

The wide variety of surgical techniques highlights the need for post-treatment regimens to be individually adapted to the procedure. However, postoperative care has been hardly researched in the field of foot surgery. There are no standardized and proven concepts for weight-bearing and the use of appropriate aids such as post-op shoes (POSs) or crutches. Typically, mobilization is carried out in a POS for 6 weeks. Besides others, the allowable postoperative load depends on the type of surgery, the osteosynthesis material used, and the quality of the bone [12]. Although POSs are widely used in postoperative care, there is no information in the instructions for use regarding the necessary gait pattern or recommended gait training. In summary, there is insufficient evidence about optimal postoperative management in terms of amount and duration of offloading, as well as partial or early full weight bearing [16].

The primary purpose of commonly used POS is to protect the surgical site regions by reducing load during walking, while repeated overloading can impair the bony consolidation and compromise soft tissue healing, potentially leading to loss of correction, malunion, or even nonunion [17]. Most POS have a rigid sole that is designed to provide protection from impact or uneven surfaces. However, this rigid sole restricts the natural rollover behavior of the anatomically and biomechanically complex structure of the human foot. In this context, there are insufficient data on the effects on forefoot loading during the now unnatural rolling phase when a patient wearing a POS attempts to roll over normally.

To encode motion information, pressure sensors have been widely used in various applications, from plantar pressure measurements to pressure-sensing floors [18,19,20,21]. Previous studies have shown that POS do not consistently reduce plantar pressure in all areas of the foot [22,23,24]. Over the past few years, various forefoot offloading shoes have been developed to provide full or partial forefoot relief using various shoe designs. While some studies have demonstrated their effectiveness in reducing mean and peak plantar pressures, comfort and pain relief remain critical factors [23,25,26]. Forefoot offloading shoes often receive lower comfort ratings, which may lead to reduced compliance and, therefore, an increased risk of complications [23,26,27,28,29].

In addition, most studies have focused on walking in POS under full load, with little attention given to partial weight-bearing or training patients in proper gait patterns preoperatively.

The aim of this study was to analyze the effects of a common POS on plantar pressure under the forefoot. It also aimed to investigate whether improper use of the POS could potentially increase pressure patterns on the forefoot.

## 2. Methodology

### 2.1. Study Population

In this study, 16 healthy adult participants without foot complaints or foot surgery in the past with an average age of 29 years (min–max: 24–39 years) took part in this study. Participants were collected on a voluntary basis via public notice at the University of Marburg. The mean height was 1.77 m (min–max: 1.67–1.96 m), and the mean body weight was 73 kg (min–max: 60–95 kg), which led to a mean body mass index (BMI) of 23.3 (min–max: 19.9–28.0). Mean foot size was 41 (min–max: 38–47).

### 2.2. Measurement of Plantar Pressure Parameters

As shown in Figure 1 pedobarographic data were obtained using the GP MulitSens^®^ plate (go-tec GmbH, Münster, Germany) and the GP MobilData^®^ insole pressure sensors (go-tec GmbH, Münster, Germany).

### 2.3. Testing Protocol

After obtaining written informed consent, the age, height, weight, and shoe size of each participant were surveyed and measured. Each participant performed three different test runs on a straight tartan track, with measurements taken once on the left foot and once on the right. The participants walked a marked distance of 10 m at a comfortable, self-selected pace, turned at a marked point, and walked back the same distance. After accelerating and before decelerating, a range of 3–5 steps in the middle of the distance was included in the measurements. Before the measurement, the participants were instructed in the correct walking technique by a certified foot surgeon and a certified sports scientist. In particular, the heel-accentuated limping gait was demonstrated, and the correct execution was monitored by the specialists present. The subjects were measured for each of the three test settings, first with their right foot and then with their left. The test setup was carried out in a defined sequence: barefoot, POS normal, and POS heel.

### 2.4. Barefoot Plantar Pressure Measurement (I)

The participant walks barefoot over an integrated measuring plate placed level with the track.

### 2.5. POS Plantar Pressure Measurement (II)

The participant wears a POS with an inserted pressure-measuring insole (Figure 1) and his normal footwear on the other foot. IIa—The test person walks normally, putting weight on the whole foot. IIb—The test person is instructed to place the weight predominantly on the heel and not to roll over the entire foot as if the forefoot had just undergone surgery. This heel-accentuated “limping” gait was instructed as a short-step gait with minimal push-off and full weight bearing.

The three walking modes are shown in Figure 2.

### 2.6. Footwear

For tests IIa and IIb, a POS (Medsurg^®^, DARCO International Inc., Huntington, CA, USA) was used, as shown in Figure 2IIa, IIb. This shoe is available in sizes S, M, L, and XL and was selected to fit each participant. The Medsurg^®^ shoe is constructed with a mild rocker design, and the rocker angle is 10°. The rocker point is 58% of the total length of the boot, while the sole is completely rigid. For this test, the shoe was only worn on one foot, while the participant wore their own footwear on the other foot to simulate conditions similar to those following unilateral forefoot surgery.

### 2.7. Data Analysis

For the statistical analysis, integrals of PP over all contact areas of the different regions were formed and divided by the specific contact area to allow a relative analysis. The plantar foot was subdivided into seven anatomical regions, based on the approach used by Fuchs and colleagues: the hallux, lesser toes, first metatarsal head (MTH1), second–third metatarsal head (MTH2–3), fourth–fifth metatarsal head (MTH4-5), midfoot, and heel [23,30]. A respective mask was applied to the data (Figure 3). The primary outcome was peak pressure ((PP) (kPa), representing the maximal pressure in a region. The PP (kPa) was calculated and averaged over all steps for each region. For the statistical analysis, integrals of the PP over the contact areas (relative PP) of the different regions were formed and divided by the contact area.

### 2.8. Statistics

Statistical analyses were performed using R (R Core Team (2013)). R: A language and environment for statistical computing. R Foundation for Statistical Computing, Vienna, Austria). Data are presented as mean ± standard deviation unless stated otherwise. Prior to this study, a G*Power analysis was performed and demonstrated a minimum required sample size of *n* = 12, assuming a large effect size (0.4) at a power of 80% and an alpha error of 5%. An ANOVA with a repeated-measures design with nonparametric tests for the main effects was used according to non-normal distributed data. Additionally, pairwise Wilcoxon rank sum tests with Bonferroni correction were used.

## 3. Results

Figure 4 and Table 1 show an analysis of the values as pressure per area. Overall, the results show that a reduction in plantar pressures in all areas of the forefoot is only achieved when using the POS with a heel-accentuated gait pattern.

Five different regions were analyzed: hallux, minor toes, MTH I, MTH II and III, and MTH IV and V. Within these regions, three different gait patterns were compared: barefoot walking, normal walking in the POS (bandage shoe normal), and heel-accentuated walking in the POS (bandage shoe). The corresponding numerical values and standard deviations are shown in Table 1.

### 3.1. Hallux

The highest relative PP in the hallux region (*p* < 0.001) was obtained during barefoot walking over the floor plate, with values of 1071.7 ± 380.7 kPa. When walking in a POS with a normal gait, the relative PP was measured at 771.1 ± 297.0 kPa. The lowest values were observed during the heel-accentuated gait, with relative PP of 141.0 ± 176.4 kPa (Table 1, Figure 4).

### 3.2. Lesser Toes

Concerning the lesser toes (*p* < 0.001), the mean relative PP during normal walking in the POS was significantly higher (467.4 ± 188.8 kPa) compared to normal barefoot walking (298.9 ± 166.0 kPa). The lowest values were observed during the heel-accentuated “limping gait” (128.2 ± 162.5 kPa) (Table 1, Figure 4).

### 3.3. MTH1

In the MTH1 area (*p* < 0.001), the highest relative PP was recorded during barefoot walking, measuring 1050.1 ± 513.6 kPa. During normal gait in a bandage shoe, the relative PP was 843.5 ± 267.5 kPa, while the lowest values were observed during the heel-accentuated gait in the bandage shoe, with a relative PP of 258.3 ± 286.6 kPa (Table 1, Figure 4).

### 3.4. MTH23

Regarding MTH23 (*p* < 0.001), the highest relative PP values of all areas were obtained when walking barefoot (2268.7 ± 369.3 kPa). This was followed by normal gait in a postop shoe (963.1 ± 194.0 kPa) and the “limping gait” in a postop shoe with the lowest values (377.6 ± 293.4 kPa) (Table 1, Figure 4).

### 3.5. MTH45

In the MTH45 region (*p* < 0.001), the relative PP values for barefoot gait (751.2 ± 441.9 kPa) and normal gait in the POS (711.4 ± 200.7 kPa) were not significantly different. As observed in all other areas, the lowest values were recorded during the heel-accentuated gait, with a pressure of 360.1 ± 282.7 kPa (Table 1, Figure 4).

### 3.6. Midfoot and Heel

For the regions of the midfoot (*p* < 0.001) and heel (*p* < 0.001), all comparisons were statistically significant except for barefoot gait (289.4 ± 133.9 kPa) and the heel-accentuated gait in the bandage shoe (316.0 ± 188.1) in the midfoot region (Table 1, Figure 4).

## 4. Discussion

The aim of this study was to analyze the effects of a common POS on plantar pressures under the forefoot and especially the way of using it. Our results demonstrate that only a heel-accentuated “limping” gait leads to a significant reduction in peak pressures in all tested areas, compared to the normal gait in a POS.

Furthermore, regarding the lesser toes, significantly higher-pressure values were found during normal walking in the POS than during barefoot walking. This indicates that a POS may not necessarily be beneficial for plantar pressures, particularly in the area of lesser toes, when not used correctly. Our results suggest that the protective function of a POS relies on its proper use, specifically the correct gait pattern.

This study was not the first to find that POS do not always have a positive impact on the plantar pressure distribution. For instance, Schuh and colleagues analyzed the plantar pressure distribution of five POS, including two models with similar characteristics to the one used in this study (Wocker^®^, Wock, Porto, Portugal, and DARCO Flat^®^, DARCO International Inc., Huntington, CA, USA) [22]. They observed high-pressure peaks in the medial forefoot area with these models, which further indicates that improper wearing of these shoes can lead to increased pressure on the forefoot, as observed in our study.

The rigid material of the sole provides low flexibility, designed to restrict movement in the vulnerable area. However, it is unclear if the stiffness of these shoes creates a considerable torque in the foot, which could lead to high pressure points in the forefoot area and potentially increase the load on the forefoot instead of reducing it [31].

Such observations were also made in a study similarly structured to ours, which focused on forefoot offloading shoes. Fuchs and colleagues examined the plantar pressure distribution of three types of these items, one rocker shoe and two wedge shoes, in healthy participants using comparable pressure measurement insoles [23].

Their results indicated that the rocker shoe did not significantly reduce pressure and even increased pressure under the lesser toes, the MTH4-5 region, and the heel. Additionally, Lorei and colleagues reported that a similar rocker shoe provided significant pressure reduction under the medial forefoot but at the cost of increased pressure on the lateral forefoot and midfoot [26]. In this context, both the angle of the rocker design and the selected tipping point also seem to play a role [32,33]. However, in our study, a standard POS was tested instead of a forefoot offloading shoe. Although no direct comparisons can be made, these studies underline the need for caution when assuming positive effects of postoperative footwear.

Additionally, Navarro-Cano and colleagues investigated the effect of applied load on plantar peak force in two postoperative shoes, including the Darco Medsurg^®^, using a biomechanical human cadaver model [24]. At a load of less than 30 kg, this study group found no difference between wearing a shoe and walking barefoot, suggesting a load threshold below which the use of an orthopedic shoe is not worthwhile. However, their study focused on pressure sensors placed under the first metatarsal head and under the heel, while the results presented here particularly examine the hallux and lesser toes, both areas that have been less studied. Our study found slightly lower PP in the area of the MTH1 when walking in a POS than when walking barefoot. However, the study by Navarro-Cano and colleagues was conducted on human cadavers where a precise and controlled vertical force was applied to the foot. In contrast, in this study presented here, it was not possible to accurately reproduce the force applied in vivo with each step.

Taken together, these studies illustrate that while certain footwear can reduce pressure in some areas of the foot, it can also shift pressure to other regions, emphasizing the need for careful selection and proper use of postoperative footwear.

Over the past few years, a significant number of forefoot offloading shoes have been designed to provide complete or partial forefoot relief using various designs. The sole type of these shoes can be categorized as either a rocker or a wedge sole [23]. Several studies show that forefoot offloading shoes are effective in reducing both midfoot and toe pressure, with wedge soles being superior to rocker soles in terms of pressure relief [23,25,26].

However, pressure reduction is not the only key factor for postoperative footwear; comfort and pain relief are essential aspects, too [28,34]. In several studies, patients rated the comfort of forefoot offloading shoes as significantly poor, with some even showing that the use of wedge shoes led to a higher incidence of back pain and higher dropout rates, particularly among older patients [23,26,27,28,29]. If the shoe causes discomfort, patients may stop wearing it, potentially raising the risk of complications. In this context, it also remains to be seen whether heel walking can result in a similar loss of comfort and possible fatigue as wearing forefoot shoes, which could also contribute to lower compliance when using POS. Further studies are also needed on the safety of heel walking in less mobile patients and the elderly.

Most of the studies conducted to date have been carried out with participants walking in a POS with full weight bearing. However, little attention has been paid to the concepts of partial weight bearing and training patients in the correct gait pattern when wearing a POS. Brown and Mueller, for example, were able to show as early as 1998 in a small study population that a gentle “step-to” gait can reduce plantar pressures in the forefoot area [35]. This was also confirmed a decade later in another working group [36].

In this study presented here, the participants were instructed to perform the heel-accentuated “limping” gait as a short-step gait with minimal push-off and full weight bearing. After all, this special training had a significantly positive effect on pressure distribution at the forefoot. In all areas tested, a significant reduction was observed with heel-emphasized walking compared to both normal walking in a POS and barefoot walking. Eidmann’s working group, which examined various POS and the influence of partial loading on the pressure distribution on the foot, also recorded a significantly reduced force ratio in the forefoot with partial loading compared to full loading for all shoes except the forefoot offloading shoe [29]. Van Schie and colleagues concluded that dedicated gait training to ensure correct use of the postoperative shoe using plantar pressure measurement is essential to optimize the pressure-reducing effect of the POS [37]. In summary, a specially trained heel-accentuated gait appears to be an effective method to further relieve the forefoot by reducing peak pressures in the respective regions. To achieve this, patients also need to be trained to walk correctly while wearing POS.

### Limitations

This study has several limitations. First, it was conducted with healthy volunteers to assess plantar pressure distribution in a POS, which may not accurately reflect the group of patients who have recently undergone forefoot surgery. However, many studies on POS have also used healthy participants [22,23,26,29,37]. Future research should aim to address this by focusing on patients in a perioperative setting and investigating a larger sample size.

In addition, the measurements were carried out on a tartan track, which does not reflect the surface on which people walk in everyday life, and the participants had a short acclimatization period for walking in the POS.

Furthermore, a major limitation of commercially available systems for measuring pressure in the sole of the foot is that they only detect forces that act perpendicular to the sensor surface [38]. Therefore, the effect of shear force on the affected areas, such as wound dehiscence, skin breakdown, or nonhealing, cannot be evaluated with these systems.

In addition, the current study design did not allow evaluation of interaction between the neuromuscular system and the POS, for example, by means of electromyography. Thus, similarly, altered pressures that may result from coactivation from the neuromuscular system postoperatively could not be examined.

Various cofounders were identified that affect the measurement of plantar force and pressure. One important factor is walking speed, as faster walking increases the vertical ground reaction forces and shifts them medially [39,40]. In this study presented here, walking speed was not standardized, as this would require the participants to focus on it, which could distort their natural gait pattern, especially since patients do not pay attention to their walking speed in real life. In addition, a slightly different implementation of heel-accentuated gait was observed among patients, but this corresponds to real-life conditions where each patient responds slightly differently to the same instruction. A potential observer bias must also be postulated as a limitation of this study design. Finally, another limiting factor is the fact that the current study is a short-term observation. The medium-term and long-term use of POS should also be investigated in the future, for example, due to the difficulty of implementation, as a simulation within the framework of a finite element analysis based on the data obtained from this study.

## 5. Conclusions

This study aimed to analyze the effects of a common POS, especially the way of using it, on plantar pressures under the forefoot compared to normal barefoot walking. The results highlight that the protective function of a POS depends on its proper use, specifically the correct gait pattern. A heel-accentuated, “limping” gait with a POS significantly reduces peak pressure values compared to both barefoot walking and walking in a POS. In contrast, normal walking in the same shoe led to significantly higher peak pressure values under the lesser toes compared to barefoot conditions. To optimize postoperative outcomes, additional strategies such as patient training in proper walking techniques should be incorporated into perioperative care. Further research based on a post-surgical study population is needed to establish clear guidelines for the ideal postoperative treatment following forefoot surgery.

## Figures and Tables

**Figure 1 bioengineering-12-00489-f001:**
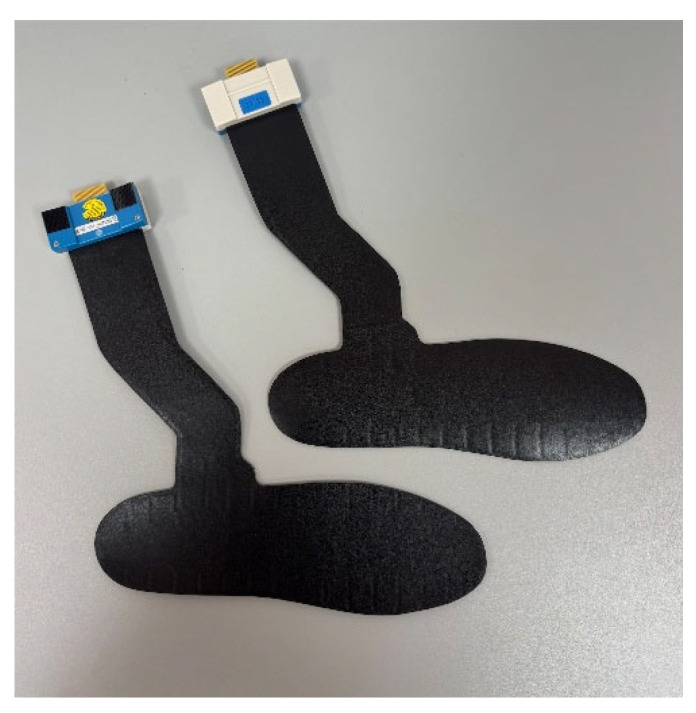
Pressure-measuring insoles.

**Figure 2 bioengineering-12-00489-f002:**
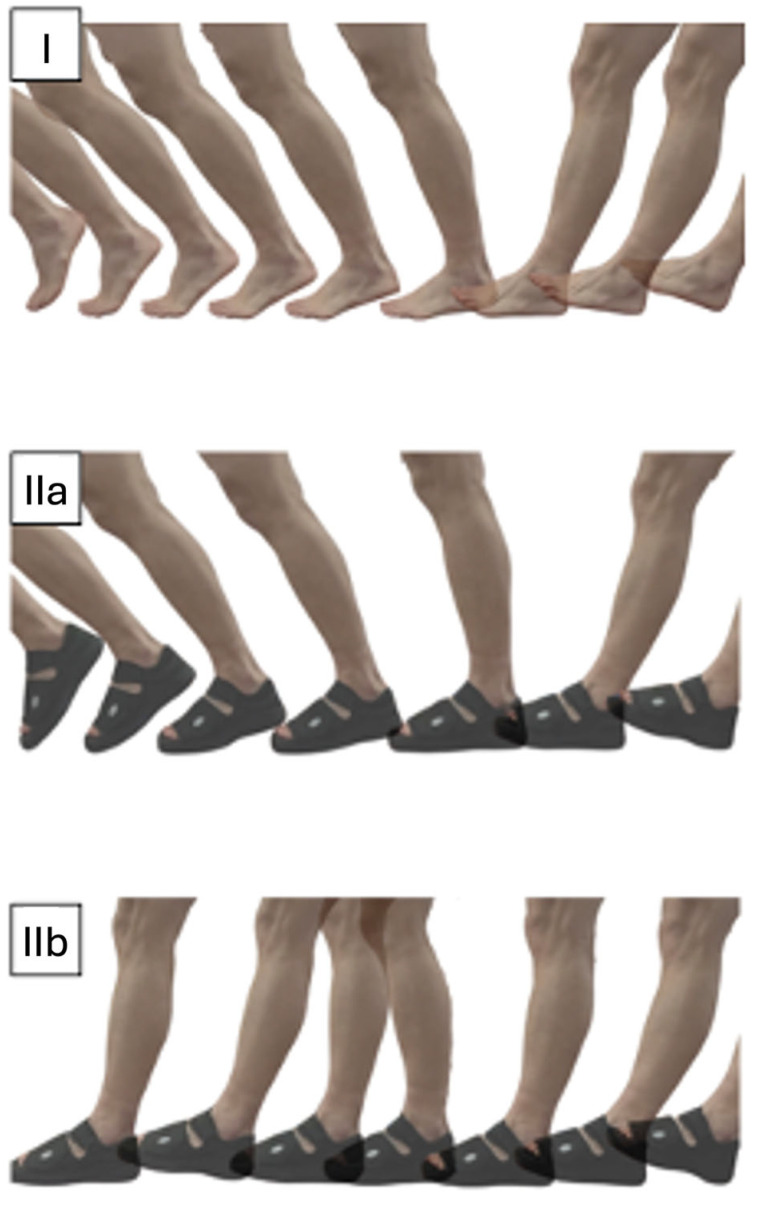
(**I**) Normal barefoot gait over a measuring plate placed on the track. (**IIa**) Normal gait in a POS with an inserted pressure measurement insole—putting weight on the whole foot. (**IIb**) Heel-accentuated “limping gait” in the POS without rolling over the entire foot.

**Figure 3 bioengineering-12-00489-f003:**
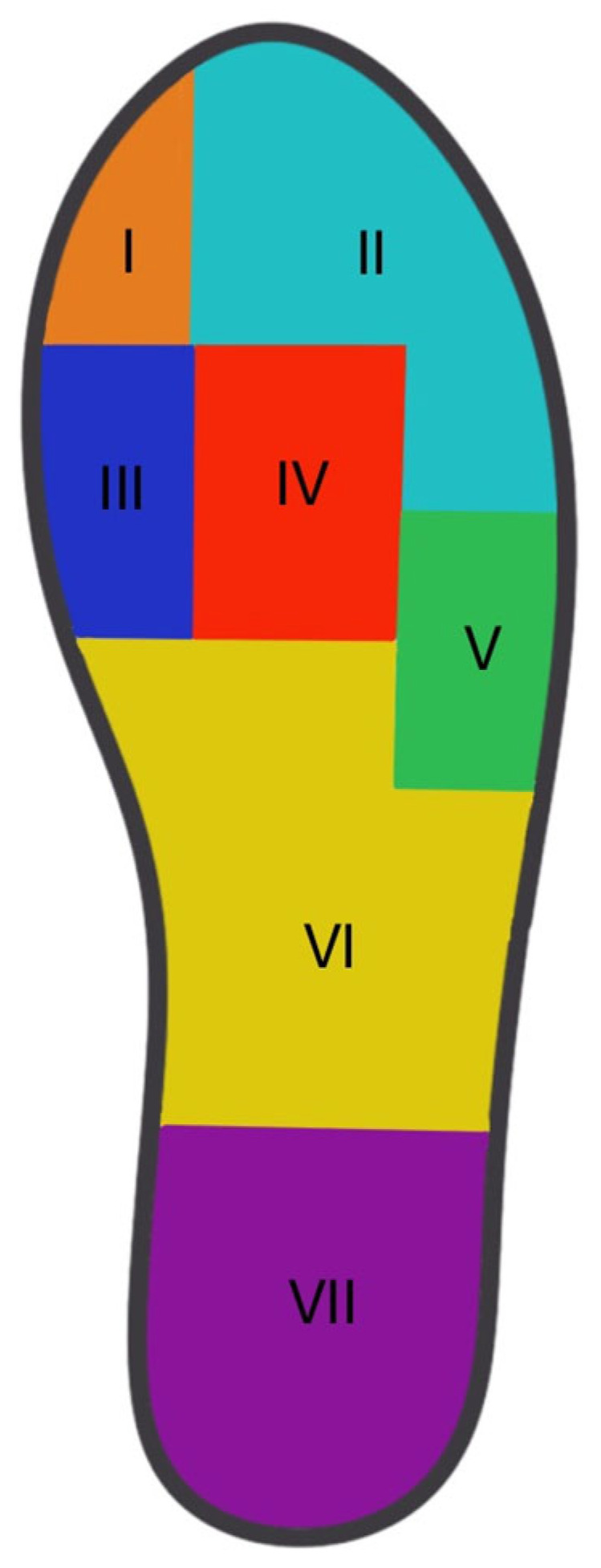
Insole division into specific regions of interest. I: Hallux, II: Lesser toes, III: MTH 1, IV: MTH 2-3, and V: MTH 4-5 (VI: Midfoot and VII: Heel are not included in this study).

**Figure 4 bioengineering-12-00489-f004:**
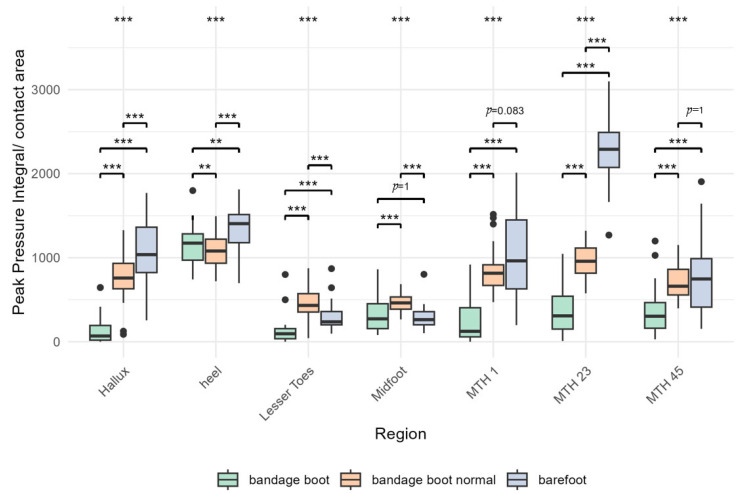
Bars illustrate the PP integral over the contact area divided by the contact area in kPa under each specific region during barefoot walking (blue), normal walking in the POS (red), and a heel-accentuated gait in the POS (green). (***: level of significance *p* < 0.001, **: *p* = 0.01, black dots: outliers).

**Table 1 bioengineering-12-00489-t001:** PP integral over the contact area/contact area ± standard deviation in kPa, ANOVA-type statistic from nonparametric ANOVA (ATS), and *p*-values for the described anatomical regions during barefoot walking (barefoot), normal walking in the POS (POS normal), and a heel-accentuated gait in the POS (POS heel).

Region	Gait Model	Mean	SD	ATS	*p*
Hallux	POS Heel POS Normal Barefoot	141.0 771.1 1071.7	176.4 297.0 380.7	114.5	*p* < 0.001
Heel	POS Heel POS Normal Barefoot	1154.8 1073.5 1353.3	231.6 199.5 252.0	15.84	*p* < 0.001
Lesser Toes	POS Heel POS Normal Barefoot	128.2 467.4 298.9	162.5 188.8 166.0	41.0	*p* < 0.001
Midfoot	POS Heel POS Normal Barefoot	316.0 468.46 289.35	188.1 111.5 133.9	14.2	*p* < 0.001
MTH1	POS Heel POS Normal Barefoot	258.3 843.5 1050.2	286.6 267.5 513.6	24.6	*p* < 0.001
MTH23	POS Heel POS Normal Barefoot	377.6 963.1 2268.7	293.4 194.0 369.3	246.1	*p* < 0.001
MTH45	POS Heel POS Normal Barefoot	360.9 711.4 751.2	282.7 200.7 441.9	15.7	*p* < 0.001

## Data Availability

The datasets used and/or analyzed during the current study are available from the corresponding author upon reasonable request.

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
