# Peer review of "Pressure-Relief Effect of Post-Op Shoes Depends on Correct Usage While Walking"

_bioengineering, 2025, doi:10.3390/bioengineering12050489_

Round 1
Reviewer 1 Report
Comments and Suggestions for Authors
The esteemed authors investigated the effect of post-operative shoes on pressure relief by analyzing insole pressure distribution data. They compared the results under three conditions. According to the presented results, some questions have been raised.
- In this study, the increased pressure on the lesser toes resulting from the use of post-operative shoes, compared to barefoot walking, has been attributed to a lack of gait training. However, the participants were healthy individuals with fully functioning motor-sensory systems. This indicates that the pressure distribution in the insoles arose from an unconscious interaction between the neuromuscular system and the shoes, aimed at maintaining dynamic balance and facilitating optimal gait dynamics. Under these circumstances, gait training was not implemented.
- An increase in insole pressure over a specific area does not necessarily indicate improper gait. The muscle coactivation patterns that determine gait dynamics are critical factors. A particular coactivation pattern developed after surgery may lead to increased pressure on an anatomical region, but this does not imply that the walking is improper.
- How can the results, obtained from healthy participants, be generalized to all injured patients suffering from various traumas and undergoing different surgical procedures? Additionally, the sample size of healthy participants is quite small.
- I recommend adding at least one reference published in 2024 to support the research.
Author Response
The esteemed authors investigated the effect of post-operative shoes on pressure relief by analyzing insole pressure distribution data. They compared the results under three conditions. According to the presented results, some questions have been raised.
- In this study, the increased pressure on the lesser toes resulting from the use of post-operative shoes, compared to barefoot walking, has been attributed to a lack of gait training. However, the participants were healthy individuals with fully functioning motor-sensory systems. This indicates that the pressure distribution in the insoles arose from an unconscious interaction between the neuromuscular system and the shoes, aimed at maintaining dynamic balance and facilitating optimal gait dynamics. Under these circumstances, gait training was not implemented.
We thank the reviewer for this valuable addition to the consideration of neuromuscular variables and fully agree that these factors also have an important influence on postoperative gait. However, this study was designed to investigate the influence of postoperative footwear and its use. Therefore, a healthy study population without too many of the above-mentioned covariates provides a relatively homogeneous basis for intraindividual comparisons. Nevertheless, we have added a statement in the limitations section of the revised manuscript to clarify the lack of real postoperative data: „In addition, the current study design did not allow evaluation of interaction between the neuromuscular system and the POS, e.g. by means of electromyography.“ (P9, LL309-312).
- An increase in insole pressure over a specific area does not necessarily indicate improper gait. The muscle coactivation patterns that determine gait dynamics are critical factors. A particular coactivation pattern developed after surgery may lead to increased pressure on an anatomical region, but this does not imply that the walking is improper.
We totally agree with the reviewer and kindly refer to our answer above. To further clarify this data gap, we have added the following sentence to the Limitations section: “Thus, similarly, altered pressures that may result from coactivation from the neuromuscular system postoperatively could not be examined.“ (P9, LL311-312).
- How can the results, obtained from healthy participants, be generalized to all injured patients suffering from various traumas and undergoing different surgical procedures? Additionally, the sample size of healthy participants is quite small.
Thank you for mentioning this important point here. Of course, it is not possible to draw conclusions about all possible identities of postoperative or posttraumatic constellations from the data of healthy study participants used here. Therefore, this limitation was included in detail in the Limitations section on Page 9, Lines 296-301. However, data from healthy subjects show a significant influence of the grossly altered gait pattern between normal gait and heel-accentuated gait. Furthermore, the power analysis carried out prior to the study is now included in the Methods section on page 5, lines 163-165 of the revised version of the manuscript and indicated a minimum required sample size of n=12: „Prior to the study a G*Power analysis was performed and demonstrated a minimum required sample size of n=12, assuming a large effect size at a power of 80%and an alpha error of 5%.“
- I recommend adding at least one reference published in 2024 to support the research.
We would like to thank the reviewer for drawing our attention to this recently published work. For this reason, we have added the reference to the relevant positions in the revised manuscript and the Reference list (Lopez A. et al (2024). The Effect of First Metatarsal Shortening and Sagittal Displacement on Forefoot Pressure in Minimally Invasive Hallux Valgus Correction. *Foot Ankle Spec*
Nouman N. et al (2024). Comparative Analysis of Three Types of Therapeutic Offloading Diabetic Shoes With Custom Made Insole on Plantar Pressure Distribution in Severe Diabetic Charcot Foot. *Can Prosthet Orthot J*).
Reviewer 2 Report
Comments and Suggestions for Authors
The research is aimed to highlight the effect that post-op shoes can cause on the pressure distribution on the foot and to analyze their effectiveness in rehabilitation after the forefoot surgeries. The study revealed that, other than shoes themselves, the gait pattern of the patient has the significant impact on the pressure distribution and peak pressure values in different foot areas which can increase the useness on the post-op shoes and can help to omit potential negative effects of the improper using of them. The research can be useful in area of the post-op shoes manufacturing and using protocols construction.
Author Response
The research is aimed to highlight the effect that post-op shoes can cause on the pressure distribution on the foot and to analyze their effectiveness in rehabilitation after the forefoot surgeries. The study revealed that, other than shoes themselves, the gait pattern of the patient has the significant impact on the pressure distribution and peak pressure values in different foot areas which can increase the useness on the post-op shoes and can help to omit potential negative effects of the improper using of them. The research can be useful in area of the post-op shoes manufacturing and using protocols construction.
Thank you very much for the favorable evaluation of our study, we also believe that the results of the presented work will be valuable for postoperative and posttraumatic protocols after foot injuries and surgical procedures.
Reviewer 3 Report
Comments and Suggestions for Authors
This paper presents a study on the impact of post-operative shoes on plantar pressure to analyze gait patterns using pressure sensors embedded in insoles and a pressure mat placed on the ground. The authors validated their approach by collecting data from 16 healthy volunteers.
I have the following major concerns in the manuscript.
1) Pressure sensors have been widely used in various applications over the past decade to encode motion information, as demonstrated in prior research [1,2]. The authors must clearly articulate their scientific contributions and highlight how their approach differs from existing studies. As it stands, the manuscript lacks significant novelty.
[1] Grosen, S. L., & Hansen, A. M. (2021). Sensor-floors: changing work and values in care for frail older persons. *Science, Technology, & Human Values*, 46(2), 254-274.
[2] Bränzel, A., et al. (2013). GravitySpace: tracking users and their poses in a smart room using a pressure-sensing floor. *Proceedings of the SIGCHI Conference on Human Factors in Computing Systems*.
2) The study primarily employs simple statistical measures such as mean and standard deviation to analyze gait patterns. More advanced techniques, particularly machine learning and deep learning algorithms, should be considered to extract meaningful gait features, as explored in [3]. This would improve the robustness and reliability of the analysis.
[3] Fatima, R., et al. (2023). A Systematic Evaluation of Feature Encoding Techniques for Gait Analysis Using Multimodal Sensory Data. *Sensors*, 24(1), 75.
3) The authors should provide graphical representations of walking patterns to illustrate both (1) inter-class variations and (2) intra-class variations. This would enhance the interpretability of the results and provide better insights into gait variations.
4) The experimental setup is too constrained, as it only includes data from healthy volunteers. To enhance the applicability of the study, the authors should consider collecting data from real patients, which would align better with the problem scenario under investigation.
5) The manuscript lacks a comparative analysis with existing state-of-the-art methods. A direct performance comparison with similar approaches in the literature would help contextualize the effectiveness of the proposed methodology.
6) The study appears more like a data collection report rather than a research article. Several major improvements are necessary for it to be considered at the journal level, including (but not limited to) the incorporation of advanced feature extraction methods, a more comprehensive experimental setup, and a clearer articulation of novel contributions.
Comments on the Quality of English LanguageMinor linguistic modification are required.
Author Response
This paper presents a study on the impact of post-operative shoes on plantar pressure to analyze gait patterns using pressure sensors embedded in insoles and a pressure mat placed on the ground. The authors validated their approach by collecting data from 16 healthy volunteers. I have the following major concerns in the manuscript.
- Pressure sensors have been widely used in various applications over the past decade to encode motion information, as demonstrated in prior research [1,2]. The authors must clearly articulate their scientific contributions and highlight how their approach differs from existing studies. As it stands, the manuscript lacks significant novelty.
[1] Grosen, S. L., & Hansen, A. M. (2021). Sensor-floors: changing work and values in care for frail older persons. *Science, Technology, & Human Values*, 46(2), 254-274.
[2] Bränzel, A., et al. (2013). GravitySpace: tracking users and their poses in a smart room using a pressure-sensing floor. *Proceedings of the SIGCHI Conference on Human Factors in Computing Systems*.
According to the reviewer suggestion, we have added the above-mentioned publications in the Introduction section: To encode motion information pressure sensors have been widely used in various applications from plantar pressure measurements to pressure-sensing floors (Grosen et al., Bränzel et al.).” (P5, L87). Nevertheless, the intention of the study presented here was not to assess the usability of foot pressure measurements with regard to different variables as in the studies by Grosen et al. and Bränzel et al. Instead, the method presented was only used to analyze the effects of a common POS on plantar pressure under the forefoot. In addition, the study hypothesis addressed the question of whether improper use of the POS could potentially increase pressure patterns on the forefoot. The clear novelty of our work is demonstrated by the conclusion that the gait pattern has a high impact on the functionality of a postoperative shoe.
- The study primarily employs simple statistical measures such as mean and standard deviation to analyze gait patterns. More advanced techniques, particularly machine learning and deep learning algorithms, should be considered to extract meaningful gait features, as explored in [3]. This would improve the robustness and reliability of the analysis.
[3] Fatima, R., et al. (2023). A Systematic Evaluation of Feature Encoding Techniques for Gait Analysis Using Multimodal Sensory Data. *Sensors*, 24(1), 75.
Thank you for this valuable information. The paper by Fatima et al. analyzes multi-modal time series sensory data and focuses on the analysis of feature encoding techniques. Machine learning models can play to their strengths and deliver very good results, especially when it comes to multimodal data. We also work with sensor data, but we do not have multimodal data, only maximum pressure as dependent variable and the experimental conditions as independent variables. Accordingly, the application of complex machine learning models seems less appropriate. Another aspect that speaks against the use of, for example, deep learning models is our theory-driven research approach. Many machine learning applications pursue the purpose of exploratory analysis or are exploratory in nature, while our research design pursues the testing of a hypothesis. This hypothesis testing should be carried out in the methodically simplest and most practical way. So, from a research-theoretical and methodological point of view, we see no necessity for the application of machine learning. However, we can well imagine using machine learning for future research projects in gait analysis.
- The authors should provide graphical representations of walking patterns to illustrate both (1) inter-class variations and (2) intra-class variations. This would enhance the interpretability of the results and provide better insights into gait variations.
Thank you for this suggestion. Unfortunately, intra-class graphical illustrations are not available and therefore cannot be added here. Nevertheless, the graphic in Figure 2, which was generated using AI supported overlay technology, illustrates the inter-class differences as well as is possible in a two-dimensional representation and thus fulfill at least one of the reviewer's suggestions.
- The experimental setup is too constrained, as it only includes data from healthy volunteers. To enhance the applicability of the study, the authors should consider collecting data from real patients, which would align better with the problem scenario under investigation.
Thank you for highlighting this important point here. Of course, it is not possible to draw conclusions about all possible identities of postoperative or posttraumatic constellations from the data of healthy study participants used here. Therefore, this limitation was included in detail in the Limitations section on Page 8, Lines 272-277. However, data from healthy subjects show a significant influence of the grossly altered gait pattern between normal gait and heel-accentuated gait.
Furthermore, we agree with the reviewer that further studies based on data from “real” patients are more than useful. However, since the ethics committee demanded the exclusive use of healthy volunteers, we are unfortunately unable to include the requested data at this time.
- The manuscript lacks a comparative analysis with existing state-of-the-art methods. A direct performance comparison with similar approaches in the literature would help contextualize the effectiveness of the proposed methodology.
The methodology used in our work corresponds to the current state of the literature with regard to in-vivo medical and clinical investigations:
- Fuchs M, Hermans MMN, Kars HJJ, Hendriks JGE, van der Steen MC. Plantar pressure distribution and wearing characteristics of three forefoot offloading shoes in healthy adult subjects. Foot (Edinb). 2020;45:101744.
- Eidmann A, Vinke W, Jakuscheit A, Rudert M, Stratos I. The influence of partial weight bearing on plantar peak forces using three different types of postoperative shoes. Foot Ankle Surg. 2022;28(8):1384-8.
- Schuh R, Trnka HJ, Sabo A, Reichel M, Kristen KH. Biomechanics of postoperative shoes: plantar pressure distribution, wearing characteristics and design criteria: a preliminary study. Arch Orthop Trauma Surg. 2011;131(2):197-203.
- Nouman M, Apiputhanayut R, Narungsri T, Tipchatyotin S, Dissaneewate T. Comparative Analysis of Three Types of Therapeutic Offloading Diabetic Shoes With Custom Made Insole on Plantar Pressure Distribution in Severe Diabetic Charcot Foot. Can Prosthet Orthot J. 2024;7(1):41780
- Carl HD, Pfander D, Swoboda B. Assessment of plantar pressure in forefoot relief shoes of different designs. Foot Ankle Int. 2006;27(2):117-20
- Lee PY, Landorf KB, Bonanno DR, Menz HB. Comparison of the pressure-relieving properties of various types of forefoot pads in older people with forefoot pain. J Foot Ankle Res. 2014;7(1):18.
The current techniques proposed by the reviewer, which we assume should be based on artificial intelligence or the studies cited with regard to GPS data and sensoring-floors, have not yet been published as state of the art in the context of clinical investigations in the field of surgery. In the future, these techniques will certainly also find their way into these areas, but ultimately, we must point out that a comprehensive ethical assessment must be consulted in clinical studies and that the use of artificial intelligence is still being critically assessed by the ethics committees.
- The study appears more like a data collection report rather than a research article. Several major improvements are necessary for it to be considered at the journal level, including (but not limited to) the incorporation of advanced feature extraction methods, a more comprehensive experimental setup, and a clearer articulation of novel contributions.
With regard to the modern methods proposed by the reviewer, we refer to the above-mentioned context. Furthermore, data acquisition in injured and postoperative patients was not possible due to ethical restrictions. The results of the represented study have a high impact on the postoperative protocols for foot surgeons worldwide. The limitations with regard to the healthy study clientele still used and the limited source of information based purely on foot pressure measurement have been extensively discussed. Articulation of the novel contributions of our work was further streamlined in the revised version of the manuscript.
Reviewer 4 Report
Comments and Suggestions for Authors
This study examined the effects of a postoperative shoe (POS) and different gait patterns on plantar pressures in the forefoot compared to barefoot walking. 16 healthy participants walked barefoot, with normal gait in a POS, and with a heel-strike gait in a POS while plantar pressure data was collected. The results showed that using a heel-strike gait with the POS significantly reduced peak pressures in the forefoot compared to barefoot walking and normal gait in the POS
The study addresses a relevant clinical question regarding optimal postoperative care for forefoot surgery patients and the paper is well written. However, the authors should consider addressing the limitations of the study in the discussion section and suggest avenues for future research to build upon these findings before this paper could be published. The methodology and the development of the POS should also be better described
General comments
- Why only analysing 5 regions while in the methods the authors presented 7 regions?
- The data are definitely not normal (SD>mean! shape of the boxplots), given the small sample size authors should considered non-parametric methods
- The authors did not standardized gait speed, this could influence the results as both spatiotemporal parameters and pressure are influenced by gait speed
- The authors should be better described gait pattern modification induce by pain and/or surgery since they only evaluate this system in young healthy individuals. The modifications may be different with age or different pain level.
- Only (really) short term effect, what about mid- or long-term modification of gait pattern with the POS? This is also crucial since patients will have to wear the system for a long period.
- In the discussion, please put your results in perspective with current literature.
Author Response
General comments
- Why only analysing 5 regions while in the methods the authors presented 7 regions?
Thank you for your interest in the other data in the analysis. As the current work deals with the pressures in the forefoot area, we have not included this data so as not to lose the focus for the reader on the basis of too much data. However, to ensure that the data is complete, the results for these two zones are available as a supplement. We have noted this accordingly in the revised manuscript: “The data of regions VI and VII are available as a supplement on request.” (P7, LL212-213).
- The data are definitely not normal (SD>mean! shape of the boxplots), given the small sample size authors should considered non-parametric methods
The review's concerns are perfectly understandable for us. However, we consider the lack of normal distribution of the data to be a minor problem because we have a balanced design. In addition, we can demonstrate very significant main effects, making a possible F-value distortion due to the lack of normal distribution less likely. Nevertheless, we calculated nonparametric repeated-measures ANOVAs for factorial designs using the nparLD package in R. The results confirmed all of our effects listed in the paper, so we recommend retaining the parametric ANOVA in order to avoid further increasing the complexity of the paper and to maintain an established methodological approach.
With regard to the seemingly small sample size, we refer to the power analysis carried out prior to the study is now included in the Methods section on page 5, lines 163-165 of the revised version of the manuscript and indicated a minimum required sample size of n=12: „Prior to the study a G*Power analysis was performed and demonstrated a minimum required sample size of n=12, assuming a large effect size at a power of 80%and an alpha error of 5%.“.
- The authors did not standardized gait speed, this could influence the results as both spatiotemporal parameters and pressure are influenced by gait speed
We fully agree with the reviewer's point. Nevertheless, a standardized gear speed can only be realized if the tests were carried out with a treadmill. As part of the design of this study, the authors conducted self-tests in various settings. Running on a treadmill at a fixed speed turned out to be too complex, as the focus here was on maintaining the correct position on the treadmill and it was less possible to perform the desired gait patterns properly. For this reason, the primary decision was made not to carry out an analysis on the treadmill. When instructing all test subjects, a speed that was as intuitively consistent as possible was required. This was not measured and corrected so as not to produce the same influencing effect of the lack of concentration on the important gait pattern.
Furthermore, the influence of gait speed on the ground reaction forces is described in the Limitations section of the revised manuscript: "One important factor is walking speed, as faster walking increases the vertical ground reaction forces and shifts them medially (35, 36)." (P9, LL314-315).
- The authors should be better described gait pattern modification induce by pain and/or surgery since they only evaluate this system in young healthy individuals. The modifications may be different with age or different pain level.
We totally agree with the reviewer that gait pattern is certainly influenced by pain or the surgical procedure itself. The patient's age is certainly also important. Nevertheless, the aim of this study was not to mimic real clinical postoperative conditions, but rather to investigate the influence of different gait patterns (when using a POS) on plantar foot pressures. Further clinical studies, also based on the data presented here, are certainly desirable in the future. In our view, therefore, the consideration of the parameters mentioned by the reviewer is not necessarily required to the current investigations.
- Only (really) short term effect, what about mid- or long-term modification of gait pattern with the POS? This is also crucial since patients will have to wear the system for a long period.
We completely agree with the reviewer on this point. Of course, it would be desirable to consider medium-term and long-term changes due to the different gaits. Such test set-ups are understandably highly complex and there are hardly any test subjects for this. It is also unclear whether the use of medical shoes over several weeks could possibly have negative consequences in otherwise healthy test subjects. We are not aware of any study in the literature that has carried out foot pressure measurements over several weeks. From our point of view, this data would certainly be desirable; for further investigations, for example, a time-dependent simulation using finite element analysis based on the data we have obtained would be conceivable. We have added this accordingly in the Limitations section: “Finally, another limiting factor is the fact that the current study is a short-term observation. The medium-term and long-term use of POS should also be investigated in the future, for example, due to the difficulty of implementation, as a simulation within the framework of a finite element analysis based on the data obtained from the study.” (P9, LL320-325).
- In the discussion, please put your results in perspective with current literature.
Thank you very much for your suggestions. Accordingly, we have rewritten several parts of the discussion section (highlighted in yellow). Nevertheless, there is actually not that much content in the literature about the influence of gait pattern upon plantar pressure while using POS. Changes were realized in the revised version of the manuscript on pages 7-9.
Reviewer 5 Report
Comments and Suggestions for Authors
Abstract:
Abstract is good and well-structured. Indicate the clinical significance of the findings. For example, make clear in the abstract: "Inappropriate usage of post-op shoes can risk increasing delayed healing or complications."
Introduction
Great background on forefoot deformities and surgical alternatives.
Brief the lengthy discussion of surgical procedures (lines 61-69) it has low relation with the topic; highlight instead lack of postop management know how, the biggest concern.
Methods
Clarification is suggested. Define how "heel-accentuated limping gait" was standardized among the participants. Were instructions, demonstrations, or observations of step cadence and heel strike given?
The study uses healthy volunteers rather than postoperative patients. While partly the rationale is described in the limitations section, such a selection reduces the clinical significance of findings. Mechanisms of gait in healthy individuals are significantly distinct from postforefoot surgical patients, particularly due to pain, swelling, altered proprioception, and sensitization of the surgery area. Above all, explicitly state why healthy subjects were used in the introduction or methodology, not just for limitations. In addition, indicate how the findings can be applied to clinical practice or propose future validation in the actual patient population.
The investigation is founded on pressure insole systems that measure only perpendicular forces and not shear forces, which are clinically significant in postoperative rehabilitation. Further explain the potential consequences of not being able to measure shear forces, in that they could lead to wound dehiscence, skin breakdown, or nonhealing.
Walking speed is not standardized; the authors state that they employed a "natural gait" for subjects. There is, however, an understanding that walking speed is a confounder for plantar pressure studies. Describe how variability in walking speed may have impacted outcome and comment on whether or not future studies would be served better by using instrumented treadmills or speed measurement to minimize variance.
Repeated-measures ANOVA corrected with Greenhouse Geisser is the appropriate statistical approach to the study design. But the sample size is small (N=16) and power analysis not mentioned. Specify if a power calculation was conducted to determine the sample size adequacy for determining significant differences.
Results
Findings are made evident with inclusive tables and figures. Include report of effect size as well as p-value for clearer communication regarding the size of difference.
Discussion
Context is established by reference to previous work. I suggest, addressing comfort and compliance explicitly within the context of heel-accentuated gait. While pressure is reduced, does this mode of gait affect patient comfort or fatigue and thus potentially lessen adherence? Elucidate whether heel-accentuated gait would be safe and possible for elderly or less mobile patients.
Limitations
Report potential observer bias if participants knew they were being watched and potentially changed their gait accordingly.
✦ Language & Formatting
Consistency: "POS" and "post-op shoe" are used interchangeably. Use one term consistently across the manuscript.
Conclusion
The research provides important information about how proper gait patterns can maximize post-op shoe protection. It is useful for patient education and postoperative training requirements.bClinical utility remains speculative in this healthy volunteer group. The study would be complemented by follow-up replication in a post-surgical patient population with standardized protocols of gait training and objective measures of compliance.
Major Revision, is suggested
Author Response
Abstract:
Abstract is good and well-structured. Indicate the clinical significance of the findings. For example, make clear in the abstract: "Inappropriate usage of post-op shoes can risk increasing delayed healing or complications."
Thank you very much for this improvement. According to the reviewer’s suggestion we have added the phrase in the revised version of the manuscript in the Abstract section: „…and possibly even increase the risk of delayed healing or complications…“ (P1, L29).
Introduction:
Great background on forefoot deformities and surgical alternatives.
Brief the lengthy discussion of surgical procedures (lines 61-69) it has low relation with the topic; highlight instead lack of postop management know-how, the biggest concern.
We fully agree with the Reviewer. For this reason, we have reworded these paragraphs in the Introduction section and shifted the focus away from surgical treatment methods to the lack of post-treatment concepts: „Available and state-of-the-art treatment options of toe deformities include conservative management and surgical techniques. More than 150 procedures are described for surgical treatment of the HV. The procedure is selected based on the severity of the hallux valgus and accompanying pathologies in the foot and ankle area (12, 13). The most common surgical techniques employed to address hammertoe deformities are resection arthroplasty, arthrodesis, and soft tissue release (14, 15). The wide variety of surgical techniques highlights the need for post-treatment regimens to be individually adapted to the procedure. However, postoperative care has been hardly researched in the field of foot surgery. There are no standardized and proven concepts for weight-bearing and the use of appropriate aids such as post-op shoes (POS) or crutches. Typically, mobilization is carried out in a POS for 6 weeks. Besides others the allowable postoperative load depends on the type of surgery, the osteosynthesis material used, and the quality of the bone (12). Although POS are widely used in postoperative care, there is no information in the instructions for use regarding the necessary gait pattern or recommended gait training. In summary, there is insufficient evidence about the optimal postoperative management in terms of amount and duration of offloading, as well as partial or early full weight bearing (16).“ (P2, LL61-77).
Methods
Clarification is suggested. Define how "heel-accentuated limping gait" was standardized among the participants. Were instructions, demonstrations, or observations of step cadence and heel strike given?
We would like to thank the reviewer for drawing our attention to this missing information in the methods section. Although the test subjects were healthy and mentally fit, it was essential that the gait types were explained and practiced in advance. For this purpose, dry exercises were carried out on the tartan track until the attending specialists and sports scientists could confirm that all subjects were sufficiently trained to ensure reproducible and comparable test sequences. To improve the manuscript accordingly, we have added an explanatory sentence to the Methods section: „Before the measurement, the participants were instructed in the correct walking technique by a certified foot surgeon and a certified sports scientist. In particular, the heel-accentuated limping gait was demonstrated, and the correct execution was monitored by the specialists present.” (P3, LL 121-124).
The study uses healthy volunteers rather than postoperative patients. While partly the rationale is described in the limitations section, such a selection reduces the clinical significance of findings. Mechanisms of gait in healthy individuals are significantly distinct from post forefoot surgical patients, particularly due to pain, swelling, altered proprioception, and sensitization of the surgery area. Above all, explicitly state why healthy subjects were used in the introduction or methodology, not just for limitations. In addition, indicate how the findings can be applied to clinical practice or propose future validation in the actual patient population.
Due to the influence of different types of gait on the pressure conditions in the foot area, which had not previously been investigated in the literature, the ethics committee's assessment stipulated that only healthy test subjects could be tested initially. This procedure can be found in many other published studies, either cadaver studies or studies on healthy test subjects.
Nevertheless, the questions raised by the reviewer are more than justified. The aim of this study was to investigate in general whether different gait patterns can influence pressure distribution in the foot area. The individual comparison with conditions that are as similar as possible appears to be very important here. We therefore believe that healthy test subjects without pain in the postoperative state can create a more compliant and reproducible test situation.
The investigation is founded on pressure insole systems that measure only perpendicular forces and not shear forces, which are clinically significant in postoperative rehabilitation. Further explain the potential consequences of not being able to measure shear forces, in that they could lead to wound dehiscence, skin breakdown, or nonhealing.
Thank you very much for this improvement. According to the reviewer’s suggestion we have added potential consequences caused by the missing shear forces measurement in the Limitations section of the revised manuscript: „Therefore, the effect of shear force on the affected areas such as wound dehiscence, skin breakdown, or nonhealing cannot be evaluated with these systems.“ (P9, LL307-308).
Walking speed is not standardized; the authors state that they employed a "natural gait" for subjects. There is, however, an understanding that walking speed is a confounder for plantar pressure studies. Describe how variability in walking speed may have impacted outcome and comment on whether or not future studies would be served better by using instrumented treadmills or speed measurement to minimize variance.
As part of the design of this study, the authors conducted self-tests in various settings. Running on a treadmill at a fixed speed turned out to be too complex, as the focus here was on maintaining the correct position on the treadmill and it was less possible to perform the desired gait patterns properly. For this reason, the primary decision was made not to carry out an analysis on the treadmill. When instructing all test subjects, a speed that was as intuitively consistent as possible was required. This was not measured and corrected so as not to produce the same influencing effect of the lack of concentration on the important gait pattern.
Furthermore, the influence of gait speed on the ground reaction forces is described in the Limitations section of the revised manuscript: "One important factor is walking speed, as faster walking increases the vertical ground reaction forces and shifts them medially (35, 36)." (P9, LL314-315).
Repeated-measures ANOVA corrected with Greenhouse Geisser is the appropriate statistical approach to the study design. But the sample size is small (N=16) and power analysis not mentioned. Specify if a power calculation was conducted to determine the sample size adequacy for determining significant differences.
The power analysis carried out prior to the study is now included in the Methods section on page 5, lines 163-165 of the revised version of the manuscript and indicated a minimum required sample size of n=12: „Prior to the study a G*Power analysis was performed and demonstrated a minimum required sample size of n=12, assuming a large effect size at a power of 80%and an alpha error of 5%.“
Results
Findings are made evident with inclusive tables and figures. Include report of effect size as well as p-value for clearer communication regarding the size of difference.
We added the generalized eta-squared η²G as a measure of effect size in ANOVA with repeated-measures design and the according p-values in the results section in the different regions in the main text as well as in Table 1 (PP6-7).
Discussion
Context is established by reference to previous work. I suggest, addressing comfort and compliance explicitly within the context of heel-accentuated gait. While pressure is reduced, does this mode of gait affect patient comfort or fatigue and thus potentially lessen adherence? Elucidate whether heel-accentuated gait would be safe and possible for elderly or less mobile patients.
We thank the reviewer for highlighting this important point. Therefore, we have added a more detailed view upon this topic in the revised version of the manuscript: “In this context, it also remains to be seen whether heel walking can result in a similar loss of comfort and possible fatigue as wearing forefoot shoes, which could also contribute to lower compliance when using POS. Further studies are also needed on the safety of heel walking in less mobile patients and the elderly.” (P8, LL273-277).
Limitations
Report potential observer bias if participants knew they were being watched and potentially changed their gait accordingly.
The point was supplemented as follows in the Limitations section: “A potential observer bias must also be postulated as a limitation of this study design." (P9, LL320-321).
Language & Formatting
Consistency: "POS" and "post-op shoe" are used interchangeably. Use one term consistently across the manuscript.
According to the reviewer’s valuable suggestion we have changed the wording to POS in the entire manuscript after the first mention.
Conclusion
The research provides important information about how proper gait patterns can maximize post-op shoe protection. It is useful for patient education and postoperative training requirements. Clinical utility remains speculative in this healthy volunteer group. The study would be complemented by follow-up replication in a post-surgical patient population with standardized protocols of gait training and objective measures of compliance.
Thank you for your valuable advice, the Conclusions section has been amended accordingly in the revised version of the manuscript: “Further research based on a post-surgical study population is needed to establish clear guidelines for the ideal postoperative treatment following forefoot surgery.“ (P9, LL335-337).
Reviewer 6 Report
Comments and Suggestions for Authors
I thank the Editor and Authors for the opportunity to read this manuscript. The authors have conducted an experimental study investigating plantar pressures in healthy volunteers walking with different shoe conditions and gait styles. The study is interesting and clinically relevant for post-operative care following foot surgery. The most part of the manuscript is easy to read and follow, but I think major restructuring is needed of the Introduction and Discussion. Please see comments below.
Title
I suggest to not say “after forefoot surgery” in the title as the participants had not gone through surgery at the time of the study.
Abstract
“This study aims to analyze the effects of a commonly used post-op shoe on plantar pressures under the forefoot and to assess whether improper usage could affect pressure patterns and potentially lead to adverse outcomes.”
I suggest deleting the latter part of the aim as the study did not investigate adverse outcomes.
Introduction
I think the introduction needs to be revised to focus more on the topic of the study (pressures on the operation site with different shoes and walking styles) and less on more peripheral topics (e.g., surgical techniques and specific toe deformities).
Line 52. “Consequences range from merely symptomatic complaints to full disability.”
Please explain the meaning of full disability or revise text.
Lines 78-82. Please add references supporting the statements in the section.
Line 81. “Most POS feature a rigid sole that provides protection from impact or uneven surfaces, but there is insufficient data on their impact on forefoot load during the rollover phase of walking.”
I do not understand the statement on protection from impact or uneven surfaces. Please explain more to improve readability.
I think there is a need for a clearer rational for the study. What knowledge gaps in the literature is this study addressing? Comfort and pain are mentioned (line 88), but these are not investigated in the study. I think the introduction needs to be more focussed on the issue here, ie, offloading effectiveness post-op being dependent on shoe configuration and walking style.
Line 95. “It also aimed to investigate whether improper use of the POS could potentially increase pressure patterns on the forefoot and lead to harmful effects.”
Please revise as the study did not investigate harmful effects.
Methods
Please describe how potential participants were approached and informed/asked for interest in participating.
Was any sample size calculation performed?
Line 100-101. I think it is enough to report the demographics on a group level, in the text. That is, I think table 1 can be deleted. Also, please specify if the values in parentheses are range (min-max) or SD.
I think the protocol for the data collection needs to be described in more detail.
E.g., were the test conditions (barefoot, POS normal, POS heel) performed in random order?
Further, the authors describe that participants walked for 10 m, turned around and then walked back, which includes acceleration and deceleration parts. Which steps (and how many) were included in the analysis?
Was the force plate in level with the rest of the walking track, or was it above the ground level?
Line 116. “Before measurement, participants were allowed to practice correct walking and partial weight bearing under instruction of the investigator.”
I think “partial weight bearing” is the same gait style that later is referred to as “limping gait” and “heel-accentuated gait”.
I suggest using a consistent term and would like to see a more detailed description. E.g., were they instructed to only partly bear weight on the foot, or only shorten the step length (to avoid push-off with the forefoot)?
I think Fig 2c suggest a step without push-off, but I think a few more frames would need to be added, as tibia is not even vertical on the last frame.
Also, I think it would be helpful to use a consistent terminology of the different conditions, in fig.2 they are labelled A, B, and C. In the text they are called I, IIa and IIb.
Was the same foot included in all measurements, and how did the authors choose between the left and right foot?
Data analysis
I agree with the 7-box approach in the analysis, but I would find it interesting to see the results for the midfoot and heel as well.
Studies often report average PP or pressure-time-integral. I am not sure why authors chose peak pressure integral / contact area. In the results, this is referred to in different ways: “mean relative PP” (line 178), “highest maximum pressures” (line 183), “highest PP” (line 188) and “relative peak pressure”. Please revise.
Results
I suggest starting with a section summarizing the big picture of the results before moving on to the results of each foot region.
Line 158-161. I think this section fits better in the Methods section.
Line 192. “In the MTH45 region, the relative peak pressure values for barefoot gait (751 ± 442) and normal gait in the post-op shoe (711.4 ± 201) were comparable.”
I suggest using a slightly more scientific terminology than “comparable”, e.g., “not significantly different”.
Tables and figures
I suggest deleting table 1 (please see above).
Table 2. What is the unit of measurement (also in Fig. 1)?
Table 2. I suggest reporting max one decimal, and use this consistently in the tables and text, to improve readability. (now it is 2 decimals in table 2, and 0-1 decimals in text)
Table 2. Change “gate” to “gait”
Fig. 4. There is substantial overlap in the bar chart between POS normal and barefoot for MTH45. Is it correct that p-value was <0.001?
Discussion
I think the Discussion section needs to be restructured to improve focus on the most relevant things to discuss, e.g. study results, comparisons with other studies, potential mechanisms explaining the results, limitations, future research, etc.
I think there are different mechanism that affect the results and that should be explained separately. First, with a “normal” gait style, the rocker sole configuration is important, and should be compared to other studies on rocker soles. I find it confusing when introducing the comparisons with wedge shoes, which have a different mechanism. Another mechanism is the special gait style that participants were instructed to use. I think this gait style needs more explanation: was axial offloading (that is, partial weight-bearing) encouraged, or only a gait style with short steps with minimal push-off? Then, these mechanism can be compared to studies that investigate similar things.
Line 213. “The rigid material of the sole provides low flexibility, designed to restrict movement in the vulnerable area. However, the stiffness of the shoe creates a considerable torque in the foot, which leads to high pressure points in the forefoot area and potentially increases the load on the forefoot instead of reducing it.”
I am not sure this is supported by evidence on other rocker bottom shoes. Please check for correctness and add references supporting the statement.
Line 225. “However, in our study, a standard double-padded POS was tested instead of a forefoot offloading shoe.”
Please explain the meaning of double-padded POS.
Line 229-240. I am not convinced of the relevance of the comparison with a cadaver study on foot pressures, at least not if the study measured static loading.
Line 244-255. I do not think discussion on the wedge sole (with a different biomechanical effect) is in line with the focus of this study.
Line 287. “In addition, a slightly different implementation of heel-strike gait was observed among patients, but this corresponds to real-life conditions where each patient responds slightly differently to the same instruction.”
Heel-strike gait is the normal gait pattern for young adults. In what way did participants implement this differently?
Conclusion
The conclusion is appropriate and reflects the study results.
Limitations
The pressures were measured on a tartan track which may not reflect the walking surface of people in daily life.
I agree that it is a limitation that the participants were not from the foot surgery population. However, this should be acknowledged as a true limitation and cannot be dismissed by that other studies have the same limitation.
I also think the short acclimation time for walking with POS, and with different gait styles, should be mentioned as a limitation.
Language, etc.
Please spell out abbreviations (e.g., PIP, PP) first time used, and then use consistently.
Please check for consistency in terms, e.g. “POS (bandage shoe normal)” is also called “POS normal”, etc.
Please check number for rounding errors, e.g. on line 174.
Comments on the Quality of English LanguageI suggest English editing of the manuscript.
Author Response
I thank the Editor and Authors for the opportunity to read this manuscript. The authors have conducted an experimental study investigating plantar pressures in healthy volunteers walking with different shoe conditions and gait styles. The study is interesting and clinically relevant for post-operative care following foot surgery. The most part of the manuscript is easy to read and follow, but I think major restructuring is needed of the Introduction and Discussion. Please see comments below.
Title
I suggest to not say “after forefoot surgery” in the title as the participants had not gone through surgery at the time of the study.
Thank you very much, we have changed the title according to your recommendation: “Pressure-relief effect of post-op shoes depends on correct usage while walking” (P1, LL2-3).
Abstract
“This study aims to analyze the effects of a commonly used post-op shoe on plantar pressures under the forefoot and to assess whether improper usage could affect pressure patterns and potentially lead to adverse outcomes.”
I suggest deleting the latter part of the aim as the study did not investigate adverse outcomes.
According to the valuable suggestion we have deleted the latter part of the above-mentioned sentence in the revised version of the manuscript: “This study aims to analyze the effects of a commonly used POS on plantar pressures under the forefoot and to assess whether improper usage could affect pressure patterns.” (P1, LL17-18).
Introduction
I think the introduction needs to be revised to focus more on the topic of the study (pressures on the operation site with different shoes and walking styles) and less on more peripheral topics (e.g., surgical techniques and specific toe deformities).
The reviewer's concern is more than justified and we fully agree. For this reason, we have reworded these paragraphs in the Introduction section and shifted the focus away from surgical treatment methods to the lack of post-treatment concepts: „Available and state-of-the-art treatment options of toe deformities include conservative management and surgical techniques. More than 150 procedures are described for surgical treatment of the HV. The procedure is selected based on the severity of the hallux valgus and accompanying pathologies in the foot and ankle area (12, 13). The most common surgical techniques employed to address hammertoe deformities are resection arthroplasty, arthrodesis, and soft tissue release (14, 15). The wide variety of surgical techniques highlights the need for post-treatment regimens to be individually adapted to the procedure. However, postoperative care has been hardly researched in the field of foot surgery. There are no standardized and proven concepts for weight-bearing and the use of appropriate aids such as post-op shoes (POS) or crutches. Typically, mobilization is carried out in a POS for 6 weeks. Besides others the allowable postoperative load depends on the type of surgery, the osteosynthesis material used, and the quality of the bone (12). Although POS are widely used in postoperative care, there is no information in the instructions for use regarding the necessary gait pattern or recommended gait training. In summary, there is insufficient evidence about the optimal postoperative management in terms of amount and duration of offloading, as well as partial or early full weight bearing (16).“ (P2, LL61-77).
Line 52. “Consequences range from merely symptomatic complaints to full disability.”
Please explain the meaning of full disability or revise text.
In order to clarify this misleading wording, we have adapted the corresponding text in the revised manuscript: ” Consequences range from merely symptomatic complaints to significant limitations in mobility and weight-bearing of the extremity.” (P2, LL52-53).
Lines 78-82. Please add references supporting the statements in the section.
According to the reviewer’s recommendation we have added the reference: “Martone J, Poel LV, Levy N. Complications of arthrodesis and nonunion. Clin Podiatr Med Surg. 2012 Jan;29(1):11-8. doi: 10.1016/j.cpm.2011.09.002. PMID: 22243566.“ (P2, L81).
Line 81. “Most POS feature a rigid sole that provides protection from impact or uneven surfaces, but there is insufficient data on their impact on forefoot load during the rollover phase of walking.”
I do not understand the statement on protection from impact or uneven surfaces. Please explain more to improve readability.
We want to thank the reviewer for detecting this misleading statement. Therefore, we have rephrased the whole text and extended the explanation for the readership: “Most POS have a rigid sole that is designed to provide protection from impact or uneven surfaces. However, the rigid sole makes it almost impossible to rollover naturally, similar to walking in ski boots. In this context, there is insufficient data on the effects on forefoot loading during the now unnatural rolling phase when a patient wearing a POS attempts to rollover normally.” (P2, LL81-85).
I think there is a need for a clearer rational for the study. What knowledge gaps in the literature is this study addressing? Comfort and pain are mentioned (line 88), but these are not investigated in the study. I think the introduction needs to be more focused on the issue here, ie, offloading effectiveness post-op being dependent on shoe configuration and walking style.
We totally agree and thank the reviewer for this improvement. Therefore, we have revised a whole paragraph of the Introduction section: “However, postoperative care has been hardly researched in the field of foot surgery. There are no standardized and proven concepts for weight-bearing and the use of appropriate aids such as post-op shoes (POS) or crutches. Typically, mobilization is carried out in a POS for 6 weeks. Besides others the allowable postoperative load depends on the type of surgery, the osteosynthesis material used, and the quality of the bone (12). Although POS are widely used in postoperative care, there is no information in the instructions for use regarding the necessary gait pattern or recommended gait training. In summary, there is insufficient evidence about the optimal postoperative management in terms of amount and duration of offloading, as well as partial or early full weight bearing (16).” (P2, LL68-77).
Line 95. “It also aimed to investigate whether improper use of the POS could potentially increase pressure patterns on the forefoot and lead to harmful effects.”
Please revise as the study did not investigate harmful effects.
According to the valuable suggestion we have deleted the latter part of the above-mentioned sentence in the revised version of the manuscript: “It also aimed to investigate whether improper use of the POS could potentially increase pressure patterns on the forefoot.” (P2, LL99-100).
Methods
Please describe how potential participants were approached and informed/asked for interest in participating.
The participants in the study were collected on a voluntary basis via a public notice at the University of Marburg.
Was any sample size calculation performed?
Yes. We have added the G*Power analysis to the Methods section in the revised manuscript: “Prior to the study a G*Power analysis was performed and demonstrated a minimum required sample size of n=12, assuming a large effect size at a power of 80% and an alpha error of 5%.“ (P5, LL163-165).
Line 100-101. I think it is enough to report the demographics on a group level, in the text. That is, I think table 1 can be deleted. Also, please specify if the values in parentheses are range (min-max) or SD.
According to the reviewer’s suggestion we have deleted Table 1 and include all missing data to the text body. In addition, we have clarified, that min-max was meant in parentheses: “16 healthy adult participants without foot complaints or foot surgery in the past with an average age of 29 years (min-max: 24-39 years) took part in this study. The mean height was 1,77m (min-max: 1,67-1,96m) and the mean bodyweight 73kg (min-max: 60-95kg). which led to a mean body mass index (BMI) of 23,3 (min-max: 19,9-28,0). Mean foot size was 41 (min-max: 38-47).” (P3, LL103-107).
I think the protocol for the data collection needs to be described in more detail.
E.g., were the test conditions (barefoot, POS normal, POS heel) performed in random order?
Further, the authors describe that participants walked for 10 m, turned around and then walked back, which includes acceleration and deceleration parts. Which steps (and how many) were included in the analysis?
According to the reviewer’s suggestion we have described the setup in a more detailed way in the revised manuscript: “After accelerating and before decelerating a range of 3-5 steps in the middle of the distance were included in the measurements. … The test setup was carried out in a defined sequence: barefoot, POS normal, POS heel.” (P3, LL119-121 & LL124-125).
Was the force plate in level with the rest of the walking track, or was it above the ground level?
The force plate was integrated in the track: “The participant walks barefoot over an integrated measuring plate placed in level with the track.” (P3, LL127-128).
Line 116. “Before measurement, participants were allowed to practice correct walking and partial weight bearing under instruction of the investigator.” I think “partial weight bearing” is the same gait style that later is referred to as “limping gait” and “heel-accentuated gait”. I suggest using a consistent term and would like to see a more detailed description. E.g., were they instructed to only partly bear weight on the foot, or only shorten the step length (to avoid push-off with the forefoot)?
In order to clarify this misleading wording, we have adapted the corresponding text in a more detailed way in the revised manuscript: “Before the measurement, the participants were instructed in the correct walking technique by a certified foot surgeon and a certified sports scientist. In particular, the heel-accentuated limping gait was demonstrated, and the correct execution was monitored by the specialists present.” (P3, LL121-124).
I think Fig 2c suggest a step without push-off, but I think a few more frames would need to be added, as tibia is not even vertical on the last frame.
Also, I think it would be helpful to use a consistent terminology of the different conditions, in fig.2 they are labelled A, B, and C. In the text they are called I, IIa and IIb.
Fig 2c shows indeed the step without push-off. It is not trivial to illustrate the gait pattern in a not animated two-dimensional image. We appreciate the reviewer’s point; however, we are not able to provide more frames. Furthermore, we totally agree and have revised the inconstant wording in Figure 2 and its legend on page 4 of the revised manuscript.
Was the same foot included in all measurements, and how did the authors choose between the left and right foot?
Both feet were observed in the study design. The subjects were measured first with their right foot and then with their left. We took this factor into account in the study design. However, for reasons of clarity and to reduce complexity, this factor was not addressed in the paper. The aspect plays only a minor role for content-related reasons for the study objective.
Data analysis
I agree with the 7-box approach in the analysis, but I would find it interesting to see the results for the midfoot and heel as well.
Thank you for your interest in the other data in the analysis. As the current work deals with the pressures in the forefoot area, we have not included this data so as not to lose the focus for the reader on the basis of too much data. However, to ensure that the data is complete, the results for these two zones is available as a supplement. We have noted this accordingly in the revised manuscript: “The data of regions VI and VII are available as a supplement on request.” (P7, LL212-213).
Studies often report average PP or pressure-time-integral. I am not sure why authors chose peak pressure integral / contact area.
In our collected data using pressure measurement soles, only the peak pressure (PP) was recorded, only the peak pressure (PP) but no time component. Both, peak pressure and the pressure-time integral are commonly used to assess plantar load. However, the pressure-time integral exhibits a strong correlation with peak pressure
(Melai T, et al, Calculation of plantar pressure time integral, an alternative approach. Gait Posture. 2011 Jul;34(3):379-83. doi: 10.1016/j.gaitpost.2011.06.005. Epub 2011 Jul 6. PMID: 21737281).
In the results, this is referred to in different ways: “mean relative PP” (line 178), “highest maximum pressures” (line 183), “highest PP” (line 188) and “relative peak pressure”. Please revise.
Maximum pressure (kPa) as an integral over the contact area divided by the contact area shows the average maximum pressure over the area. We have now standardized this as “relative PP”.
Results
I suggest starting with a section summarizing the big picture of the results before moving on to the results of each foot region.
According to the reviewer’s suggestion we have started summarizing the big picture of the results in the revised version of manuscript before moving on to the results of each foot region: “Overall, the results show that a reduction in plantar pressures in all areas of the forefoot is only achieved when using the POS with a heel-accentuated gait pattern.” (P5, LL170-172).
Line 158-161. I think this section fits better in the Methods section.
According to the reviewer’s suggestion we have placed this statement in the Data analysis paragraph of the Methods section.
Line 192. “In the MTH45 region, the relative peak pressure values for barefoot gait (751 ± 442) and normal gait in the post-op shoe (711.4 ± 201) were comparable.”
I suggest using a slightly more scientific terminology than “comparable”, e.g., “not significantly different”.
According to the reviewer’s suggestion we have rephrased this statement in the revised manuscript: “In the MTH45 region, the relative peak pressure values for barefoot gait (751 ± 442) and normal gait in the POS (711.4 ± 201) were not significantly different.” (P7, LL209-210).
Tables and figures
I suggest deleting table 1 (please see above).
Table 1 was deleted accordingly.
Table 2. What is the unit of measurement (also in Fig. 1)?
Maximum pressure (kPa) as an integral over the contact area divided by the contact area results in kPa. We have inserted the unit in Table 1 and Fig. 1.
Table 2. I suggest reporting max one decimal, and use this consistently in the tables and text, to improve readability. (now it is 2 decimals in table 2, and 0-1 decimals in text)
Table 2. Change “gate” to “gait”
The requested adjustments and changes were made accordingly (PP6-7).
Fig. 4. There is substantial overlap in the bar chart between POS normal and barefoot for MTH45. Is it correct that p-value was <0.001?
Thank you for this very attentive comment. We would like to apologize for this mistake. Of course, there is no significant difference between barefoot and POS normal in the MTH45 area with this large overlap. To avoid further transmission errors, we have re-analyzed all post-hoc tests and compared them with the figure. No further errors could be found.
Discussion
I think the Discussion section needs to be restructured to improve focus on the most relevant things to discuss, e.g. study results, comparisons with other studies, potential mechanisms explaining the results, limitations, future research, etc.
In line with the reviewer's suggestion, we have made changes to the discussion section to meet the requirements to improve focus on the most relevant aspects to discuss.
I think there are different mechanism that affect the results and that should be explained separately. First, with a “normal” gait style, the rocker sole configuration is important, and should be compared to other studies on rocker soles. I find it confusing when introducing the comparisons with wedge shoes, which have a different mechanism.
We fully agree that there are different mechanisms when talking about FOS and POS. Nevertheless, most of the publications that had a major impact on our study evaluate both entities. The changes that were now made in the Discussion section on page 8 of the revised manuscript (highlighted in yellow) improve the understanding of the discussion section.
Another mechanism is the special gait style that participants were instructed to use. I think this gait style needs more explanation: was axial offloading (that is, partial weight-bearing) encouraged, or only a gait style with short steps with minimal push-off? Then, these mechanism can be compared to studies that investigate similar things.
We want to thank the reviewer for detecting this misleading statement. We have rephrased the paragraphs in the Discussion section and extended the explanation of the gait style: “In the study presented here, the participants were instructed to perform the heel-strike “limping” gait as a short-step gait with minimal push-off. After all, this special training had a significantly positive effect on pressure distribution at the forefoot.” and “In summary, a specially trained heel-accentuated gait appears to be an effective method to further relieve the forefoot by reducing peak pressures in the respective regions.” (P8-9, LL281-283 & LL291-293).
Line 213. “The rigid material of the sole provides low flexibility, designed to restrict movement in the vulnerable area. However, the stiffness of the shoe creates a considerable torque in the foot, which could lead to high pressure points in the forefoot area and potentially increases the load on the forefoot instead of reducing it.”
I am not sure this is supported by evidence on other rocker bottom shoes. Please check for correctness and add references supporting the statement.
Thank you for reviewing our statement in detail. We agree that it is a bit bold and have therefore corrected our presentation and added an additional reference that does not directly refer to rocker shoes but explains our thoughts on the subject: “However, it’s unclear if the stiffness of these shoes creates a considerable torque in the foot, which could lead to high pressure points in the forefoot area and potentially increases the load on the forefoot instead of reducing it (31).” (P7, LL233-235).
Line 225. “However, in our study, a standard double-padded POS was tested instead of a forefoot offloading shoe.”
Please explain the meaning of double-padded POS.
Thank you for drawing our attention to this confusing wording. We have therefore deleted “double-padded” from the respective sentence in the revised version of the manuscript (P8, L244).
Line 229-240. I am not convinced of the relevance of the comparison with a cadaver study on foot pressures, at least not if the study measured static loading.
We agree with the reviewer. Of course, comparability with a cadaver study is not possible, at least not in absolute numbers. However, the study group around Navarro-Cano worked with the same POS as we do. In addition, we consider the similar trends to be a constellation that is definitely worth mentioning and therefore wish to leave this passage in the discussion.
Line 244-255. I do not think discussion on the wedge sole (with a different biomechanical effect) is in line with the focus of this study.
At first glance, the discussion of the various shoe models used in the field of foot surgery does not correspond to the original focus of this study. Nevertheless, we believe that it is worth discussing these shoes in this context, as it may also be useful for the industry to deal with this topic in various ways for future studies or shoe developments. For this reason, we would like to leave this part of the discussion open.
Line 287. “In addition, a slightly different implementation of heel-strike gait was observed among patients, but this corresponds to real-life conditions where each patient responds slightly differently to the same instruction.”
Heel-strike gait is the normal gait pattern for young adults. In what way did participants implement this differently?
We agree with the reviewer. Heel walking should not be a problem for healthy young people. Here, we wanted to emphasize that the heel walk looked slightly different in the eyes of the spectators. This is certainly an individual fact that results from a person's individual gait. The statement we made here in the Limitations obviously creates more confusion than it contributes to the manuscript in a meaningful way. For this reason, we have deleted this passage.
Conclusion
The conclusion is appropriate and reflects the study results.
Thank you.
Limitations
The pressures were measured on a tartan track which may not reflect the walking surface of people in daily life.
I agree that it is a limitation that the participants were not from the foot surgery population. However, this should be acknowledged as a true limitation and cannot be dismissed by that other studies have the same limitation.
I also think the short acclimation time for walking with POS, and with different gait styles, should be mentioned as a limitation.
We agree with the reviewer and have revised the Limitations section according to the suggestions made by the reviewer: “In addition, the measurements were carried out on a tartan track, which does not reflect the surface on which people walk in everyday life, and the participants had a short acclimatization period for walking in the POS.” (P9, LL302-304).
Language, etc.
Please spell out abbreviations (e.g., PIP, PP) first time used, and then use consistently.
Please check for consistency in terms, e.g. “POS (bandage shoe normal)” is also called “POS normal”, etc.
Please check number for rounding errors, e.g. on line 174.
We thank the Reviewer for making us aware of these points and have made the respective corrections and adjustments in the revised version of the manuscript.
Round 2
Reviewer 1 Report
Comments and Suggestions for Authors
I want to thank the authors for addressing all the points raised.
Author Response
I want to thank the authors for addressing all the points raised.
We thank you for your valuable improvements.
Reviewer 4 Report
Comments and Suggestions for Authors
The authors successfully managed to answer most of the point and improved the quality of the paper, especially the clinical part.
However the authors still used wrong statistics: the data are not normally distributed (see boxplots and huge SD in comparison with mean). The authors should therefore used non-parametric statistics (or at least robust method).
Author Response
The authors successfully managed to answer most of the point and improved the quality of the paper, especially the clinical part.
However, the authors still used wrong statistics: the data are not normally distributed (see boxplots and huge SD in comparison with mean). The authors should therefore used non-parametric statistics (or at least robust method).
After another careful revision of the statistical methodology including expert contribution of professional statisticians we disagree with this point. We used a balanced design reducing the relevance of the lack of normal distribution of the data for analysis. We can demonstrate very significant main effects, making a possible F-value distortion due to the lack of normal distribution less likely. Nevertheless, we calculated nonparametric repeated-measures ANOVAs for factorial designs using the nparLD package in R. The results confirmed all effects listed in the paper, so we recommend retaining the parametric ANOVA in order to avoid further increasing the complexity of the paper and to maintain an established methodological approach.
Reviewer 6 Report
Comments and Suggestions for Authors
I thank the authors for having addressed most of my concerns and questions in their revision of the manuscript. However, some concerns remain, please see below.
Introduction
I appreciate that the authors have changed some of the focus in the introduction with less emphasis on surgery and more emphasis on the topic for the study. Still, approximately half of the introduction concerns foot deformities and surgery, and I believe this could be condensed to further to sharpen the focus on the topic, that is, how to offload post-operatively.
I think I now understand the statement that “Most POS feature a rigid sole that provides protection from impact or uneven surfaces” but I would discourage the comparison with ski boots as they have different biomechanical features than POS shoes: they also immobilize the ankle joint and have a rigid flat (non-rocker) sole.
Methods and Results
I thank the authors for having clarified how participants were approached and informed/asked for interest in participating (“The participants in the study were collected on a voluntary basis via a public notice at the University of Marburg”). Please also add the description to the manuscript text.
I thank the authors for the clarification of the sample size calculation. Please define “large effect size” in numbers.
I appreciate the elaboration of the gait style that the participants were instructed to use. However, I suggest to mainly describe this in the Methods, not Discussion. Please also clarify if partial or full weight-bearing was used on the “limping” limb.
I think there is a need for a description of the POS used. Is the sole completely rigid? It looks like a forefoot rocker -please describe, and if possible quantify the rocker sole’s properties/parameters (rocker angle, position, etc.).
Was the same foot included in all measurements, and how did the authors choose between the left and right foot?
Both feet were observed in the study design. The subjects were measured first with their right foot and then with their left. We took this factor into account in the study design. However, for reasons of clarity and to reduce complexity, this factor was not addressed in the paper. The aspect plays only a minor role for content-related reasons for the study objective.
I strongly suggest that this is added to the manuscript.
Data analysis
I agree with the 7-box approach in the analysis, but I would find it interesting to see the results for the midfoot and heel as well.
Thank you for your interest in the other data in the analysis. As the current work deals with the pressures in the forefoot area, we have not included this data so as not to lose the focus for the reader on the basis of too much data. However, to ensure that the data is complete, the results for these two zones is available as a supplement. We have noted this accordingly in the revised manuscript: “The data of regions VI and VII are available as a supplement on request.” (P7, LL212-213).
The Results section is rather short at present, so I think there is room for adding the results of the midfoot and heel. This would help readers better understand the effects of the shoe and gait style, e.g., if the heel accentuated gait resulted in higher loads on the heel.
I am still not sure why peak pressure integral / contact area was calculated, as opposed to average PP which I think is more commonly reported. Please provide a rational for this. This may need to be reviewed by someone with more expertise in pressure measurement. Also, I do not understand the term “relative PP”, what is the PP relativized to?
Discussion
I still think the Discussion section needs major revisions, to find a structure that focus on the mechanisms under investigation in the current study, that is, shoe design and gait style. There are many studies on rocker sole design, and I think the results of the current study need to be discussed and interpreted in this context. Similarly, gait style changes should be compared with other studies (if available) investigating this. I still think aspects outside the study (e.g., wedge soles) should be mentioned only briefly, if at all, to not distract from the focus of the Discussion.
Line 213. “The rigid material of the sole provides low flexibility, designed to restrict movement in the vulnerable area. However, the stiffness of the shoe creates a considerable torque in the foot, which could lead to high pressure points in the forefoot area and potentially increases the load on the forefoot instead of reducing it.”
I am not sure this is supported by evidence on other rocker bottom shoes. Please check for correctness and add references supporting the statement.
Thank you for reviewing our statement in detail. We agree that it is a bit bold and have therefore corrected our presentation and added an additional reference that does not directly refer to rocker shoes but explains our thoughts on the subject: “However, it’s unclear if the stiffness of these shoes creates a considerable torque in the foot, which could lead to high pressure points in the forefoot area and potentially increases the load on the forefoot instead of reducing it (31).” (P7, LL233-235).
I agree that the effects of rocker soles may vary, depending on rocker sole parameters (rocker position and angle, etc.), and I think this deserves more elaboration when comparing the study results to the results from other studies.
Please remove the term “double-padded” from line 141 as well.
Line 229-240. I am not convinced of the relevance of the comparison with a cadaver study on foot pressures, at least not if the study measured static loading.
We agree with the reviewer. Of course, comparability with a cadaver study is not possible, at least not in absolute numbers. However, the study group around Navarro-Cano worked with the same POS as we do. In addition, we consider the similar trends to be a constellation that is definitely worth mentioning and therefore wish to leave this passage in the discussion.
I understand the authors’ position, but I think more description of the study setup is needed to improve understanding. Was is a static or dynamic setup in the cadaver study? Only 30 kg load was used, which seems very little, if not meant to simulate standing on both feet.
Line 287. “In addition, a slightly different implementation of heel-strike gait was observed among patients, but this corresponds to real-life conditions where each patient responds slightly differently to the same instruction.”
Heel-strike gait is the normal gait pattern for young adults. In what way did participants implement this differently?
We agree with the reviewer. Heel walking should not be a problem for healthy young people. Here, we wanted to emphasize that the heel walk looked slightly different in the eyes of the spectators. This is certainly an individual fact that results from a person's individual gait. The statement we made here in the Limitations obviously creates more confusion than it contributes to the manuscript in a meaningful way. For this reason, we have deleted this passage.
I think the term “heel-strike gait” is incorrect (as heel-strike gait is the normal gait style), I believe the authors are referring to the heel accentuated gait style which was implemented in the study.
Author Response
I thank the authors for having addressed most of my concerns and questions in their revision of the manuscript. However, some concerns remain, please see below.
Introduction
I appreciate that the authors have changed some of the focus in the introduction with less emphasis on surgery and more emphasis on the topic for the study. Still, approximately half of the introduction concerns foot deformities and surgery, and I believe this could be condensed to further to sharpen the focus on the topic, that is, how to offload post-operatively.
Thank you for your assessment. As the Journal on Bioengineering is an interdisciplinary journal and there are probably many readers without a surgical background, the background content should not be missing.
I think I now understand the statement that “Most POS feature a rigid sole that provides protection from impact or uneven surfaces” but I would discourage the comparison with ski boots as they have different biomechanical features than POS shoes: they also immobilize the ankle joint and have a rigid flat (non-rocker) sole.
We share the review's assessment 100%. Nevertheless, we would like to retain this comparison for readers who may not be familiar with the topic. A comparison with a common everyday object also clearly explains to non-medical professionals what it is all about. Details such as the rocker sole or different levels of stiffness are considered in too much detail in this context.
Methods and Results
I thank the authors for having clarified how participants were approached and informed/asked for interest in participating (“The participants in the study were collected on a voluntary basis via a public notice at the University of Marburg”). Please also add the description to the manuscript text.
According to the reviewer’s suggestion, we have added the missing information to the Methods section of the revised manuscript: “Participants were collected on a voluntary basis via a public notice at the University of Marburg” (P3, LL104-105).
I thank the authors for the clarification of the sample size calculation. Please define “large effect size” in numbers.
We used Cohen's f with the standard effect sizes (0.1 = small, 0.25 = medium, 0.4 = large), which are also suggested by the software G*Power used. Therefore, we have added the effect size in numbers to the revised version of the manuscript on page 5, line 170.
I appreciate the elaboration of the gait style that the participants were instructed to use. However, I suggest to mainly describe this in the Methods, not Discussion. Please also clarify if partial or full weight-bearing was used on the “limping” limb.
Many thanks for this valuable improvement. We have introduced the corresponding passage in the method: This heel-accentuated “limping” gait was instructed as a short-step gait with minimal push-off and full weight bearing.“ (P3, LL136-137). In addition, it was also clarified in the discussion section that this was a full weight bearing (P8, L287-288).
I think there is a need for a description of the POS used. Is the sole completely rigid? It looks like a forefoot rocker -please describe, and if possible, quantify the rocker sole’s properties/parameters (rocker angle, position, etc.).
We would like to thank the reviewer for this improvement. Accordingly, we have added the parameters of the shoe used in the methods section of our revised manuscript on page 4, lines 146-147: “The Medsurg® shoe is constructed with a mild rocker design and the rocker angle is 10°. The rocker point is 58% of the total length of the boot, while the sole is completely rigid.”
Was the same foot included in all measurements, and how did the authors choose between the left and right foot? Both feet were observed in the study design. The subjects were measured first with their right foot and then with their left. We took this factor into account in the study design. However, for reasons of clarity and to reduce complexity, this factor was not addressed in the paper. The aspect plays only a minor role for content-related reasons for the study objective. I strongly suggest that this is added to the manuscript.
According to the reviewer’s suggestion, we have added the missing information to the Methods section of the revised manuscript: “The subjects were measured first with their right foot and then with their left.” (P3, LL125-126).
Data analysis
The Results section is rather short at present, so I think there is room for adding the results of the midfoot and heel. This would help readers better understand the effects of the shoe and gait style, e.g., if the heel accentuated gait resulted in higher loads on the heel.
We appreciate the Reviewer´s interest in further data of the analysis. As the current work deals with the pressures in the forefoot area, we have not included this data so as not to lose the focus for the reader on the basis of too much data. However, to ensure that the data is complete, the results for these two zones is available as a supplement. We have noted this accordingly in the revised manuscript: “The data of regions VI and VII are available as a supplement on request.” (P7, LL217-218).
I am still not sure why peak pressure integral / contact area was calculated, as opposed to average PP which I think is more commonly reported. Please provide a rational for this. This may need to be reviewed by someone with more expertise in pressure measurement. Also, I do not understand the term “relative PP”, what is the PP relativized to?
As the literature shows, the used method is very well established and common. As visualized in the point pressure integral per contact area, this is the relation between pressure and contact area. We have described the term relative PP the Data analysis subsection on page 4 of the revised manuscript.
Discussion
I still think the Discussion section needs major revisions, to find a structure that focus on the mechanisms under investigation in the current study, that is, shoe design and gait style. There are many studies on rocker sole design, and I think the results of the current study need to be discussed and interpreted in this context. Similarly, gait style changes should be compared with other studies (if available) investigating this. I still think aspects outside the study (e.g., wedge soles) should be mentioned only briefly, if at all, to not distract from the focus of the Discussion.
Thank you for another detailed review of our discussion. We cannot fully agree. The aim of our study is not to compare different shoe models. Rather, we would like to bring the correct use of these aids into the focus of science. As already explained, there is no data available in the literature on this topic, particularly on the corresponding gait pattern and training for using the POS aid. We would therefore like to retain the discussion we have rewritten in the revised version.
I agree that the effects of rocker soles may vary, depending on rocker sole parameters (rocker position and angle, etc.), and I think this deserves more elaboration when comparing the study results to the results from other studies.
The study did not intend to make a comparative analysis of different rocker designs, but merely to show that the gait pattern has an impact when using the same shoe. We agree that further studies with comparative analyses of different shoes would be useful, but this is not the subject of the manuscript presented here.
Please remove the term “double-padded” from line 141 as well.
Thank you, we have removed the term in the revised manuscript in accordance with your comment.
Line 229-240. I am not convinced of the relevance of the comparison with a cadaver study on foot pressures, at least not if the study measured static loading.
We agree with the reviewer. Of course, comparability with a cadaver study is not possible, at least not in absolute numbers. However, the study group around Navarro-Cano worked with the same POS as we do. In addition, we consider the similar trends to be a constellation that is definitely worth mentioning and therefore wish to leave this passage in the discussion.
I understand the authors’ position, but I think more description of the study setup is needed to improve understanding. Was it a static or dynamic setup in the cadaver study? Only 30 kg load was used, which seems very little, if not meant to simulate standing on both feet.
As mentioned before, we believe that a more detailed description of the study protocol of a cited study would seem overloaded for the scope of the discussion. As already confirmed in our first response, we know that the comparison with a cadaver study is not possible in absolute terms. Nevertheless, the trend from this analysis should also be included in the discussion. We believe that we have been able to round off the relevant discussion on the intended topic well with this valuable study and would like to leave it in the manuscript without adding any complicating explanations.
Line 287. “In addition, a slightly different implementation of heel-strike gait was observed among patients, but this corresponds to real-life conditions where each patient responds slightly differently to the same instruction.”
Heel-strike gait is the normal gait pattern for young adults. In what way did participants implement this differently?
We agree with the reviewer. Heel walking should not be a problem for healthy young people. Here, we wanted to emphasize that the heel walk looked slightly different in the eyes of the spectators. This is certainly an individual fact that results from a person's individual gait. The statement we made here in the Limitations obviously creates more confusion than it contributes to the manuscript in a meaningful way. For this reason, we have deleted this passage.
I think the term “heel-strike gait” is incorrect (as heel-strike gait is the normal gait style), I believe the authors are referring to the heel accentuated gait style which was implemented in the study.
Thank you very much, we apologize for this mistake, of course we meant heel-accentuated. We have corrected this accordingly in the revised manuscript: P5, L180; P8, L287; P9, LL323-324.
Round 3
Reviewer 4 Report
Comments and Suggestions for Authors
If the results are (quasi) similar with non-parametric method, authors should present these results and not parametric ones (again given the small sample size and the distribution of the data).
I leave the final decision to the editor.
Good luck with this submission
Author Response
If the results are (quasi) similar with non-parametric method, authors should present these results and not parametric ones (again given the small sample size and the distribution of the data). I leave the final decision to the editor. Good luck with this submission.
Thank you for your comment. We take the review's concerns about the statistical analysis very seriously. In round one and round two of the review process, we presented our view of things. Nevertheless, we are of course following the reviewer's suggestions and have adjusted the analyses as recommended. Therefore, we have now implemented the nonparametric ANOVA and nonparametric post hoc tests in the paper. We use the nparld package in R with the ANOVA type statistic ATS, which is particularly suitable for small sample sizes. For the post hoc tests, we used the Wilcoxon rank sum test. For the nonparametric ANOVA, no classic effect sizes are specified as eta-squares, but only relative treatment effects (RTEs), which must be analyzed across the different factor levels. Since this analysis is very complex, the eta-squared column in Table 1 on page 6 has been omitted and we have decided not to analyze the RTEs. The tables and graphs have been adjusted based on the changes: „An ANOVA with a repeated-measures design with nonparametric tests for the main effects was used according to non-normal distributed data. Additionally, pairwise Wilcoxon rank sum tests with Bonferroni correction were used.“ (P5, LL158-160).
Reviewer 6 Report
Comments and Suggestions for Authors
I thank the authors for having addressed some of my concerns and questions. However, some major concerns remain, and if not solved, I cannot recommend this manuscript for publication. I have summed them up under each heading, please see below.
Introduction
I agree that the journal and its readers are interdisciplinary, and I think this is a further reason to focus and explain the topic of the study in more depth, that is, offloading post-operatively with POS and gait style changes. Prevalence of foot deformities and surgical techniques are secondary in this context, which I think the introduction should reflect.
I understand the purpose of the comparison with ski boots but disagree that it is appropriate: walking with a ski boot (stiff sole, flat sole, immobilized ankle), is very different to walking with a stiff rocker-bottom post-op shoe. If an everyday comparison is needed, I suggest comparing with clog shoe or something similar, which is closer to a post-op shoe in its features. I also disagree with the statement “However, the rigid sole makes it almost impossible to rollover naturally…” as rigid (rocker) soles are used in, e.g., hiking boots.
Methods
I appreciate the clarifications in the last revision, but still do not fully understand the protocol for tests and analysis. Did each subject walk under each condition twice, that is, first measuring pressures under the right foot then the left foot (for barefoot, POS normal, POS heel)? Were the average of the right and left foot analyzed and presented in the results?
Results
I strongly suggest adding the results of the midfoot and heel pressures, as this sheds lights on the effects of the POS and heel gait style on the entire foot. I do not see any reason to exclude this from the results presentation or to ask readers to contact the authors if interested in these results.
Discussion
I still think the Discussion section needs major revision to find a clearer structure with more focus on contextualizing the findings in the research literature. There are studies available on post-operative shoes with rocker soles, gait style changes (e.g., Brown & Mueller (1998). A “step-to” gait decreases pressures on the forefoot. Journal of Orthopaedic & Sports Physical Therapy, 28(3), 139-145), and several studies on rocker shoes (not being POS, but with similar mechanical features). My point is not that authors should to discuss specific shoe models, but the mechanisms of action that were investigated in the current study.
Author Response
I thank the authors for having addressed some of my concerns and questions. However, some major concerns remain, and if not solved, I cannot recommend this manuscript for publication. I have summed them up under each heading, please see below.
Introduction
I agree that the journal and its readers are interdisciplinary, and I think this is a further reason to focus and explain the topic of the study in more depth, that is, offloading post-operatively with POS and gait style changes. Prevalence of foot deformities and surgical techniques are secondary in this context, which I think the introduction should reflect.
According to the reviewer’s suggestion we have compressed the clinical part of the Introduction section (PP1-2, L36-51). Additionally, explanation of the study’s focus was brought to the fore (P2, LL53-71).
I understand the purpose of the comparison with ski boots but disagree that it is appropriate: walking with a ski boot (stiff sole, flat sole, immobilized ankle), is very different to walking with a stiff rocker-bottom post-op shoe. If an everyday comparison is needed, I suggest comparing with clog shoe or something similar, which is closer to a post-op shoe in its features. I also disagree with the statement “However, the rigid sole makes it almost impossible to rollover naturally…” as rigid (rocker) soles are used in, e.g., hiking boots.
According to the reviewer valuable suggestion we have rephrased the above-mentioned statements: “However, this rigid sole restricts the natural rollover behavior of the anatomically and biomechanically complex structure of the human foot.” (P2, L68-70).
Methods
I appreciate the clarifications in the last revision but still do not fully understand the protocol for tests and analysis. Did each subject walk under each condition twice, that is, first measuring pressures under the right foot then the left foot (for barefoot, POS normal, POS heel)? Was the average of the right and left foot analyzed and presented in the results?
Thank you. The test subjects were measured once with each of the three test settings on their right foot and once with their left foot. Accordingly, the left/right factor was considered in the calculation. To reduce complexity, we decided not to include the right/left factor in the results section. We can see slightly higher pressure on the right side in some regions, but this does not affect the central findings of the paper. Therefore, we have explained the test protocol more detailed now: “The subjects were measured for each of the three test settings first with their right foot and then with their left.” (P3, LL112-113). Unfortunately, we did not carry out a differentiated analysis of the sides, so that no average values can be presented in the results for the right and left feet.
Results
I strongly suggest adding the results of the midfoot and heel pressures, as this sheds lights on the effects of the POS and heel gait style on the entire foot. I do not see any reason to exclude this from the results presentation or to ask readers to contact the authors if interested in these results.
According to the reviewer’s suggestion we have added the missing results in the text on page 7, lines 207-209: “For the regions midfoot (p<0.001) and heel (p<0.001), all comparisons were statistically significant except for barefoot gait (289.4 ± 133.9 kPa) and the heel-accentuated gait in the bandage shoe (316.0 ± 188.1) in the Midfoot region (Table 1, Fig. 4).”, in Table 1 on pages 6-7 and in Figure 4 on page 6 of the revised version of the manuscript.
Discussion
I still think the Discussion section needs major revision to find a clearer structure with more focus on contextualizing the findings in the research literature. There are studies available on post-operative shoes with rocker soles, gait style changes (e.g., Brown & Mueller (1998). A “step-to” gait decreases pressures on the forefoot. Journal of Orthopaedic & Sports Physical Therapy, 28(3), 139-145), and several studies on rocker shoes (not being POS, but with similar mechanical features). My point is not that authors should discuss specific shoe models, but the mechanisms of action that were investigated in the current study.
According to the reviewer’s suggestions we have rewritten several parts of the discussion and have implemented the suggested reference and several more references as in the Discussion section of the revised manuscript: “In this context, both the angle of the rocker design and the selected tipping point also seem to play a role (Preece SJ, Chapman JD, Braunstein B, Brüggemann GP, Nester CJ. Optimisation of rocker sole footwear for prevention of first plantar ulcer: comparison of group-optimised and individually-selected footwear designs. J Foot Ankle Res. 2017 Jul 6;10:27. doi: 10.1186/s13047-017-0208-3. PMID: 28694849; PMCID: PMC5501571.)” (P8, LL240-241) and “Brown and Mueller, for example, were able to show as early as 1998 in a small study population that a gentle “step-to” gait can reduce plantar pressures in the forefoot area (Brown HE, Mueller MJ. A "step-to" gait decreases pressures on the forefoot. J Orthop Sports Phys Ther. 1998 Sep;28(3):139-45. doi: 10.2519/jospt.1998.28.3.139. PMID: 9742470.). This was also confirmed a decade later in another working group (Drerup B, Szczepaniak A, Wetz HH. Plantar pressure reduction in step-to gait: a biomechanical investigation and clinical feasibility study. Clin Biomech (Bristol). 2008 Oct;23(8):1073-9. doi: 10.1016/j.clinbiomech.2008.04.014. Epub 2008 Jun 13. PMID: 18555568.).” (P8-9, LL279-281).